# Robust and Communication-Efficient Collaborative Learning

**Amirhossein Reisizadeh**
ECE Department
University of California, Santa Barbara
reisizadeh@ucsb.edu

**Hossein Taheri**
ECE Department
University of California, Santa Barbara
hossein@ucsb.edu

**Aryan Mokhtari**
ECE Department
The University of Texas at Austin
mokhtari@austin.utexas.edu

**Hamed Hassani**
ESE Department
University of Pennsylvania
hassani@seas.upenn.edu

**Ramtin Pedarsani**
ECE Department
University of California, Santa Barbara
ramtin@ece.ucsb.edu

## Abstract

We consider a decentralized learning problem, where a set of computing nodes aim at solving a non-convex optimization problem collaboratively. It is well-known that decentralized optimization schemes face two major system bottlenecks: stragglers' delay and communication overhead. In this paper, we tackle these bottlenecks by proposing a novel decentralized and gradient-based optimization algorithm named as `QuanTimed-DSGD`. Our algorithm stands on two main ideas: (i) we impose a *deadline* on the local gradient computations of each node at each iteration of the algorithm, and (ii) the nodes exchange *quantized* versions of their local models. The first idea robustifies to straggling nodes and the second alleviates communication efficiency. The key technical contribution of our work is to prove that with non-vanishing noises for quantization and stochastic gradients, the proposed method *exactly* converges to the global optimal for convex loss functions, and finds a first-order stationary point in non-convex scenarios. Our numerical evaluations of the `QuanTimed-DSGD` on training benchmark datasets, MNIST and CIFAR-10, demonstrate speedups of up to $3\times$ in run-time, compared to state-of-the-art decentralized optimization methods.

## 1   Introduction

Collaborative learning refers to the task of learning a common objective among multiple computing agents without any central node and by using on-device computation and local communication among neighboring agents. Such tasks have recently gained considerable attention in the context of machine learning and optimization as they are foundational to several computing paradigms such as scalability to larger datasets and systems, data locality, ownership and privacy. As such, collaborative learning naturally arises in various applications such as distributed deep learning (LeCun et al., 2015; Dean et al., 2012), multi-agent robotics and path planning (Choi and How, 2010; Jha et al., 2016), distributed resource allocation in wireless networks (Ribeiro, 2010), to name a few.

While collaborative learning has recently drawn significant attention due its decentralized implementation, it faces major challenges at the system level as well as algorithm design. The decentralized implementation of collaborative learning faces two major systems challenges: (i) significant slow-down due to straggling nodes, where a subset of nodes can be largely delayed in their local computation which slows down the wall-clock time convergence of the decentralized algorithm; (ii) large communication overhead due to the message passing algorithm as the dimension of the parameter vector increases, which can further slow down the algorithm's convergence time. Moreover, in the presence of these system bottlenecks, the efficacy of classical consensus optimization methods is not clear and needs to be revisited.

In this work we consider the general data-parallel setting where the data is distributed across different computing nodes, and develop decentralized optimization methods that do not rely on a central coordinator but instead only require local computation and communication among neighboring nodes. As the main contribution of this paper, we propose a *straggler-robust* and *communication-efficient* algorithm for collaborative learning called `QuanTimed-DSGD`, which is a quantized and deadline-based decentralized stochastic gradient descent method. We show that the proposed scheme provably improves upon on the convergence time of vanilla synchronous decentralized optimization methods. The key theoretical contribution of the paper is to develop the *first* quantized decentralized non-convex optimization algorithm with provable and *exact* convergence to a first-order optimal solution.

There are two key ideas in our proposed algorithm. To provide robustness against stragglers, we impose a *deadline* time $T_d$ for the computation of each node. In a synchronous implementation of the proposed algorithm, at every iteration all the nodes simultaneously start computing stochastic gradients by randomly picking data points from their local batches and evaluating the gradient function on the picked data point. By $T_d$, each node has computed a random number of stochastic gradients from which it aggregates and generates a stochastic gradient for its local objective. By doing so, each iteration takes a constant computation time as opposed to deadline-free methods in which each node has to wait for all their neighbours to complete their gradient computation tasks. To tackle the communication bottleneck in collaborative learning, we only allow the decentralized nodes to share with neighbours a *quantized* version of their local models. Quantizing the exchanged models reduces the communication load which is critical for large and dense networks.

We analyze the convergence of the proposed `QuanTimed-DSGD` for strongly convex and non-convex loss functions and under standard assumptions for the network, quantizer and stochastic gradients. In the strongly convex case, we show that `QuanTimed-DSGD` *exactly* finds the global optimal for *every* node with a rate arbitrarily close to $\mathcal{O}(1/\sqrt{T})$. In the non-convex setting, `QuanTimed-DSGD` provably finds first-order optimal solutions as fast as $\mathcal{O}(T^{-1/3})$. Moreover, the consensus error decays with the same rate which guarantees an exact convergence by choosing large enough $T$. Furthermore, we numerically evaluate `QuanTimed-DSGD` on benchmark datasets CIFAR-10 and MNIST, where it demonstrates speedups of up to $3\times$ in the run-time compared to state-of-the-art baselines.

**Related Work.** Decentralized consensus optimization has been studied extensively. The most popular first-order choices for the convex setting are distributed gradient descent-type methods (Nedic and Ozdaglar, 2009; Jakovetic et al., 2014; Yuan et al., 2016; Qu and Li, 2017), augmented Lagrangian algorithms (Shi et al., 2015a,b; Mokhtari and Ribeiro, 2016), distributed variants of the alternating direction method of multipliers (ADMM) (Schizas et al., 2008; Boyd et al., 2011; Shi et al., 2014; Chang et al., 2015; Mokhtari et al., 2016), dual averaging (Duchi et al., 2012; Tsianos et al., 2012), and several dual based strategies (Seaman et al., 2017; Scaman et al., 2018; Uribe et al., 2018). Recently, there have been some works which study non-convex decentralized consensus optimization and establish convergence to a stationary point (Zeng and Yin, 2018; Hong et al., 2017, 2018; Sun and Hong, 2018; Scutari et al., 2017; Scutari and Sun, 2018; Jiang et al., 2017; Lian et al., 2017a).

The idea of improving communication-efficiency of distributed optimization procedures via message-compression schemes goes a few decades back (Tsitsiklis and Luo, 1987), however, it has recently gained considerable attention due to the growing importance of distributed applications. In particular, efficient gradient-compression methods are provided in (Alistarh et al., 2017; Seide et al., 2014; Bernstein et al., 2018) and deployed in the distributed master-worker setting. In the decentralized setting, quantization methods were proposed in different convex optimization contexts with *non-vanishing* errors (Yuksel and Basar, 2003; Rabbat and Nowak, 2005; Kashyap et al., 2006; El Chamie et al., 2016; Aysal et al., 2007; Nedic et al., 2008). The first *exact* decentralized optimization method with quantized messages was given in (Reisizadeh et al., 2018; Zhang et al., 2018), and more

recently, new techniques have been developed in this context for convex problems (Doan et al., 2018; Koloskova et al., 2019; Berahas et al., 2019; Lee et al., 2018a,b).

The straggler problem has been widely observed in distributed computing clusters (Dean and Barroso, 2013; Ananthanarayanan et al., 2010). A common approach to mitigate stragglers is to replicate the computing task of the slow nodes to other computing nodes (Ananthanarayanan et al., 2013; Wang et al., 2014), but this is clearly not feasible in collaborative learning. Another line of work proposed using coding theoretic ideas for speeding up distributed machine learning (Lee et al., 2018c; Tandon et al., 2016; Yu et al., 2017; Reisizadeh et al., 2019b,c), but they work mostly for master-worker setup and particular computation types such as linear computations or full gradient aggregation. The closest work to ours is (Ferdinand et al., 2019) that considers decentralized optimization for convex functions with deadline for local computations without considering communication bottlenecks and quantization as well as non-convex functions. Another line of work proposes asynchronous decentralized SGD, where the workers update their models based on the last iterates received by their neighbors (Recht et al., 2011; Lian et al., 2017b; Lan and Zhou, 2018; Peng et al., 2016; Wu et al., 2017; Dutta et al., 2018). While asynchronous methods are inherently robust to stragglers, they can suffer from slow convergence due to using stale models.

## 2 Problem Setup

In this paper, we focus on a stochastic learning model in which we aim to solve the problem

$$\min_{\mathbf{x}} L(\mathbf{x}) := \min_{\mathbf{x}} \mathbb{E}_{\theta \sim \mathcal{P}}[\ell(\mathbf{x}, \theta)], \tag{1}$$

where $\ell : \mathbb{R}^p \times \mathbb{R}^q \to \mathbb{R}$ is a stochastic loss function, $\mathbf{x} \in \mathbb{R}^p$ is our optimization variable, and $\theta \in \mathbb{R}^q$ is a random variable with probability distribution $\mathcal{P}$ and $L : \mathbb{R}^p \to \mathbb{R}$ is the expected loss function also called population risk. We assume that the underlying distribution $\mathcal{P}$ of the random variable $\theta$ is unknown and we have access only to $N = mn$ realizations of it. Our goal is to solve the loss associated with $N = mn$ realizations of the random variable $\theta$, which is also known as empirical risk minimization. To be more precise, we aim to solve the empirical risk minimization (ERM) problem

$$\min_{\mathbf{x}} L_N(\mathbf{x}) := \min_{\mathbf{x}} \frac{1}{N} \sum_{k=1}^{N} \ell(\mathbf{x}, \theta_k), \tag{2}$$

where $L_N$ is the empirical loss associated with the sample of random variables $\mathcal{D} = \{\theta_1, \ldots, \theta_N\}$.

**Collaborative Learning Perspective**. Our goal is to solve the ERM problem in (2) in a decentralized manner over $n$ nodes. This setting arises in a plethora of applications where either the total number of samples $N$ is massive and data cannot be stored or processed over a single node or the samples are available in parts at different nodes and, due to privacy or communication constraints, exchanging raw data points is not possible among the nodes. Hence, we assume that each node $i$ has access to $m$ samples and its local objective is

$$f_i(\mathbf{x}) = \frac{1}{m} \sum_{j=1}^{m} \ell(\mathbf{x}, \theta_i^j), \tag{3}$$

where $\mathcal{D}_i = \{\theta_i^1, \cdots, \theta_i^m\}$ is the set of samples available at node $i$. Nodes aim to collaboratively minimize the average of all local objective functions, denoted by $f$, which is given by

$$\min_{\mathbf{x}} f(\mathbf{x}) = \min_{\mathbf{x}} \frac{1}{n} \sum_{i=1}^{n} f_i(\mathbf{x}) = \min_{\mathbf{x}} \frac{1}{mn} \sum_{i=1}^{n} \sum_{j=1}^{m} \ell(\mathbf{x}, \theta_i^j). \tag{4}$$

Indeed, the objective functions $f$ and $L_N$ are equivalent if $\mathcal{D} := \mathcal{D}_1 \cup \cdots \cup \mathcal{D}_n$. Therefore, by minimizing the global objective function $f$ we also obtain the solution of the ERM problem in (2).

We can rewrite the optimization problem in (4) as a classical decentralized optimization problem as follows. Let $\mathbf{x}_i$ be the decision variable of node $i$. Then, (4) is equivalent to

$$\min_{\mathbf{x}_1, \ldots, \mathbf{x}_n} \frac{1}{n} \sum_{i=1}^{n} f_i(\mathbf{x}_i), \qquad \text{subject to} \quad \mathbf{x}_1 = \cdots = \mathbf{x}_n, \tag{5}$$

as the objective function value of (4) and (5) are the same when the iterates of all nodes are the same and we have *consensus*. The challenge in distributed learning is to solve the global loss only by exchanging information with neighboring nodes and ensuring that nodes' variables stay close to each other. We consider a network of computing nodes characterized by an undirected connected graph $\mathcal{G} = (\mathcal{V}, \mathcal{E})$ with nodes $\mathcal{V} = [n] = \{1, \cdots, n\}$ and edges $\mathcal{E} \subseteq \mathcal{V} \times \mathcal{V}$, and each node $i$ is allowed to exchange information only with its neighboring nodes in the graph $\mathcal{G}$, which we denote by $\mathcal{N}_i$.

In a stochastic optimization setting, where the true objective is defined as an expectation, there is a limit to the accuracy with which we can minimize $L(\mathbf{x})$ given only $N = nm$ samples, even if we have access to the optimal solution of the empirical risk $L_N$. In particular, it has been shown that when the loss function $\ell$ is convex, the difference between the population risk $L$ and the empirical risk $L_N$ corresponding to $N = mn$ samples with high probability is uniformly bounded by $\sup_{\mathbf{x}} |L(\mathbf{x}) - L_N(\mathbf{x})| \leq \mathcal{O}(1/\sqrt{N}) = \mathcal{O}(1/\sqrt{nm})$; see (Bottou and Bousquet, 2008). Thus, without collaboration, each node can minimize its local cost $f_i$ to reach an estimate for the optimal solution with an error of $\mathcal{O}(1/\sqrt{m})$. By minimizing the aggregate loss collaboratively, nodes reach an approximate solution of the expected risk problem with a smaller error of $\mathcal{O}(1/\sqrt{nm})$. Based on this formulation, our goal in the convex setting is to find a point $\mathbf{x}_i$ for each node $i$ that attains the statistical accuracy, i.e., $\mathbb{E}\left[L_N(\mathbf{x}_i) - L_N(\hat{\mathbf{x}}^*)\right] \leq \mathcal{O}(1/\sqrt{mn})$, which further implies $\mathbb{E}\left[L(\mathbf{x}_i) - L(\mathbf{x}^*)\right] \leq \mathcal{O}(1/\sqrt{mn})$.

For a non-convex loss function $\ell$, however, $L_N$ is also non-convex and solving the problem in (4) is hard, in general. Therefore, we only focus on finding a point that satisfies the first-order optimality condition for (4) up to some accuracy $\rho$, i.e., finding a point $\tilde{\mathbf{x}}$ such that $\|\nabla L_N(\tilde{\mathbf{x}})\| = \|\nabla f(\tilde{\mathbf{x}})\| \leq \rho$. Under the assumption that the gradient of loss is sub-Gaussian, it has been shown that with high probability the gap between the gradients of expected risk and empirical risk is bounded by $\sup_{\mathbf{x}} \|\nabla L(\mathbf{x}) - \nabla L_N(\mathbf{x})\|_2 \leq \mathcal{O}(1/\sqrt{nm})$; see (Mei et al., 2018). As in the convex setting, by solving the aggregate loss instead of local loss, each node finds a better approximate for a first-order stationary point of the expected risk $L$. Therefore, our goal in the non-convex setting is to find a point that satisfies $\|\nabla L_N(\mathbf{x})\| \leq \mathcal{O}(1/\sqrt{mn})$ which also implies $\|\nabla L(\mathbf{x})\| \leq \mathcal{O}(1/\sqrt{mn})$.

## 3   Proposed `QuanTimed-DSGD` Method

In this section, we present our proposed `QuanTimed-DSGD` algorithm that takes into account robustness to stragglers and communication efficiency in decentralized optimization. To ensure robustness to stragglers' delay, we introduce a *deadline-based* protocol for updating the iterates in which nodes compute their local gradients estimation only for a specific amount time and then use their gradient estimates to update their iterates. This is in contrast to the mini-batch setting, in which nodes have to wait for the slowest machine to finish its local gradient computation. To reduce the communication load, we assume that nodes only exchange a quantized version of their local iterates. However, using quantized messages induces extra noise in the decision making process which makes the analysis of our algorithm more challenging. A detailed description of the proposed algorithm is as follows.

**Deadline-Based Gradient Computation.** Consider the current model $\mathbf{x}_{i,t}$ available at node $i$ at iteration $t$. Recall the definition of the local objective function $f_i$ at node $i$ defined in (3). The cost of computing the local gradient $\nabla f_i$ scales linearly by the number of samples $m$ assigned to the $i$-th node. A common solution to reduce the computation cost at each node for the case that $m$ is large is using a mini-batch approximate of the gradient, i.e., each node $i$ picks a subset of its local samples $\mathcal{B}_{i,t} \subseteq \mathcal{D}_i$ to compute the stochastic gradient $\frac{1}{|\mathcal{B}_{i,t}|} \sum_{\theta \in \mathcal{B}_{i,t}} \nabla \ell(\mathbf{x}_{i,t}, \theta)$. A major challenge for this procedure is the presence of stragglers in the network: given mini-batch size $b$, *all* nodes have to compute the average of exactly $b$ stochastic gradients. Thus, all the nodes have to wait for the *slowest* machine to finish its computation and exchange its new model with the neighbors.

To resolve this issue, we propose a deadline-based approach in which we set a fixed deadline $T_d$ for the time that each node can spend computing its local stochastic gradient estimate. Once the deadline is reached, nodes find their gradient estimate using whatever computation (mini-batch size) they could perform. Thus, with this deadline-based procedure, nodes do not need to wait for the slowest machine to update their iterates. However, their mini-batch size and consequently the noise of their gradient approximation will be different. To be more specific, let $\mathcal{S}_{i,t} \subseteq \mathcal{D}_i$ denote the set of random

---

**Algorithm 1** `QuanTimed-DSGD` at node $i$

---

**Require:** Weights $\{w_{ij}\}_{j=1}^{n}$, total iterations $T$, deadline $T_d$
1: Set $\mathbf{x}_{i,0} = 0$ and compute $\mathbf{z}_{i,0} = Q(\mathbf{x}_{i,0})$
2: **for** $t = 0, \cdots, T - 1$ **do**
3:      Send $\mathbf{z}_{i,t} = Q(\mathbf{x}_{i,t})$ to $j \in \mathcal{N}_i$ and receive $\mathbf{z}_{j,t}$
4:      Pick and evaluate stochastic gradients $\{\nabla \ell(\mathbf{x}_{i,t}; \theta) : \theta \in \mathcal{S}_{i,t}\}$ till reaching the deadline $T_d$
       and generate $\widetilde{\nabla} f_i(\mathbf{x}_{i,t})$ according to (6)
5:      Update $\mathbf{x}_{i,t+1}$ as follows: $\mathbf{x}_{i,t+1} = (1 - \varepsilon + \varepsilon w_{ii})\mathbf{x}_{i,t} + \varepsilon \sum_{j \in \mathcal{N}_i} w_{ij} \mathbf{z}_{j,t} - \alpha \varepsilon \widetilde{\nabla} f_i(\mathbf{x}_{i,t})$
6: **end for**

---

samples chosen at time $t$ by node $i$. Define $\widetilde{\nabla} f_i(\mathbf{x}_{i,t})$ as the stochastic gradient of node $i$ at time $t$ as

$$\widetilde{\nabla} f_i(\mathbf{x}_{i,t}) = \frac{1}{|\mathcal{S}_{i,t}|} \sum_{\theta \in \mathcal{S}_{i,t}} \nabla \ell(\mathbf{x}_{i,t}; \theta), \tag{6}$$

for $1 \leq |\mathcal{S}_{i,t}| \leq m$. If there are not any gradients computed by $T_d$, i.e., $|\mathcal{S}_{i,t}| = 0$, we set $\widetilde{\nabla} f_i(\mathbf{x}_{i,t}) = 0$.

**Computation Model.** To illustrate the advantage of our deadline-based scheme over the fixed mini-batch scheme, we formally state the model that we use for the processing time of nodes in the network. We remark that our algorithms are oblivious to the choice of the computation model which is merely used for analysis. We define the processing speed of each machine as the number of stochastic gradients $\nabla \ell(\mathbf{x}, \theta)$ that it computes per second. We assume that the processing speed of each machine $i$ and iteration $t$ is a random variable $V_{i,t}$, and $V_{i,t}$'s are i.i.d. with probability distribution $F_V(v)$. We further assume that the domain of the random variable $V$ is bounded and its realizations are in $[\underline{v}, \bar{v}]$. If $V_{i,t}$ is the number of stochastic gradient which can be computed per second, the size of mini-batch $\mathcal{S}_{i,t}$ is a random variable given by $|\mathcal{S}_{i,t}| = V_{i,t} T_d$.

In the fixed mini-batch scheme and for any iteration $t$, all the nodes have to wait for the machine with the slowest processing time before updating their iterates, and thus the overall computation time will be $b/V_{\min}$ where $V_{\min}$ is defined as $V_{\min} = \min\{V_{1,t}, \ldots, V_{n,t}\}$. In our deadline-based scheme there is a fixed deadline $T_d$ which limits the computation time of the nodes, and is chosen such that $T_d = \mathbb{E}\left[b/V\right] = b\mathbb{E}\left[1/V\right]$, while the mini-batch scheme requires an expected time of $\mathbb{E}\left[b/V_{\min}\right] = b\mathbb{E}\left[1/V_{\min}\right]$. The gap between $\mathbb{E}\left[1/V\right]$ and $\mathbb{E}\left[1/V_{\min}\right]$ depends on the distribution of $V$, and can be unbounded in general growing with $n$.

**Quantized Message-Passing.** To reduce the communication overhead of exchanging variables between nodes, we use quantization schemes that significantly reduces the required number of bits. More precisely, instead of sending $\mathbf{x}_{i,t}$, the $i$-th node sends $\mathbf{z}_{i,t} = Q(\mathbf{x}_{i,t})$ which is a quantized version of its local variable $\mathbf{x}_{i,t}$ to its neighbors $j \in \mathcal{N}_i$. As an example, consider the low precision quantizer specified by scale factor $\eta$ and $s$ bits with the representable range $\{-\eta \cdot 2^{s-1}, \cdots, -\eta, 0, \eta, \cdots, \eta \cdot (2^s - 1)\}$. For any $k\eta \leq x < (k+1)\eta$, the quantizer outputs

$$Q_{(\eta,b)}(x) = \begin{cases} k\eta & \text{w.p. } 1 - (x - k\eta)/\eta, \\ (k+1)\eta & \text{w.p. } (x - k\eta)/\eta. \end{cases} \tag{7}$$

**Algorithm Update.** Once the local variables are exchanged between neighboring nodes, each node $i$ uses its local stochastic gradient $\widetilde{\nabla} f_i(\mathbf{x}_{i,t})$, its local decision variable $\mathbf{x}_{i,t}$, and the information received from its neighbors $\{\mathbf{z}_{j,t} = Q(\mathbf{x}_{j,t}); j \in \mathcal{N}_i\}$ to update its local decision variable. Before formally stating the update of `QuanTimed-DSGD`, let us define $w_{ij}$ as the weight that node $i$ assigns to the information that it receive from node $j$. If $i$ and $j$ are not neighbors $w_{ij} = 0$. These weights are considered for averaging over the local decision variable $x_{i,t}$ and the quantized variables $\mathbf{z}_{j,t}$ received from neighbors to enforce consensus among neighboring nodes. Specifically, at time $t$, node $i$ updates its decision variable according to the update

$$\mathbf{x}_{i,t+1} = (1 - \varepsilon + \varepsilon w_{ii})\mathbf{x}_{i,t} + \varepsilon \sum_{j \in \mathcal{N}_i} w_{ij} \mathbf{z}_{j,t} - \alpha \varepsilon \widetilde{\nabla} f_i(\mathbf{x}_{i,t}), \tag{8}$$

where $\alpha$ and $\varepsilon$ are positive scalars that behave as stepsize. Note that the update in (8) shows that the updated iterate is a linear combination of the weighted average of node $i$'s neighbors' decision

variable, i.e., $\varepsilon \sum_{j \in \mathcal{N}_i} w_{ij} \mathbf{z}_{j,t}$, and its local variable $\mathbf{x}_{i,t}$ and stochastic gradient $\widetilde{\nabla} f_i(\mathbf{x}_{i,t})$. The parameter $\alpha$ behaves as the stepsize of the gradient descent step with respect to local objective function and the parameter $\varepsilon$ behaves as an averaging parameter between performing the distributed gradient update $\varepsilon(w_{ii}\mathbf{x}_{i,t} + \sum_{j \in \mathcal{N}_i} w_{ij}\mathbf{z}_{j,t} - \alpha \widetilde{\nabla} f_i(\mathbf{x}_{i,t}))$ and using the previous decision variable $(1-\varepsilon)\mathbf{x}_{i,t}$. By choosing a diminishing stepsize $\alpha$ we control the noise of stochastic gradient evaluation, and by averaging using the parameter $\varepsilon$ we control randomness induced by exchanging quantized variables. The description of `QuanTimed-DSGD` is summarized in Algorithm 1.

## 4 Convergence Analysis

In this section, we provide the main theoretical results for the proposed `QuanTimed-DSGD` algorithm. We first consider strongly convex loss functions and characterize the convergence rate of `QuanTimed-DSGD` for achieving the global optimal solution to the problem (4). Then, we focus on the non-convex setting and show that the iterates generated by `QuanTimed-DSGD` find a stationary point of the cost in (4) while the local models are close to each other and the consensus constraint is asymptotically satisfied. All the proofs are provided in the supplementary material (Section 6). We make the following assumptions on the weight matrix, the quantizer, and local objective functions.

**Assumption 1.** The weight matrix $W \in \mathbb{R}^{n \times n}$ with entries $w_{ij} \geq 0$ satisfies the following conditions: $W = W^\top$, $W\mathbf{1} = \mathbf{1}$ and $\text{null}(I - W) = \text{span}(\mathbf{1})$.

**Assumption 2.** The random quantizer $Q(\cdot)$ is unbiased and variance-bounded, i.e., $\mathbb{E}[Q(\mathbf{x})|\mathbf{x}] = \mathbf{x}$ and $\mathbb{E}[\|Q(\mathbf{x}) - \mathbf{x}\|^2|\mathbf{x}] \leq \sigma^2$, for any $\mathbf{x} \in \mathbb{R}^p$; and quantizations are carried out independently.

Assumption 1 implies that $W$ is symmetric and doubly stochastic. Moreover, all the eigenvalues of $W$ are in $(-1, 1]$, i.e., $1 = \lambda_1(W) \geq \lambda_2(W) \geq \cdots \geq \lambda_n(W) > -1$ (e.g. (Yuan et al., 2016)). We also denote by $1 - \beta$ the spectral gap associated with the stochastic matrix $W$, where $\beta = \max\{|\lambda_2(W)|, |\lambda_n(W)|\}$.

**Assumption 3.** The function $\ell$ is $K$-smooth with respect to $\mathbf{x}$, i.e., for any $\mathbf{x}, \hat{\mathbf{x}} \in \mathbb{R}^p$ and any $\theta \in \mathcal{D}$, $\|\nabla\ell(\mathbf{x}, \theta) - \nabla\ell(\hat{\mathbf{x}}, \theta)\| \leq K\|\mathbf{x} - \hat{\mathbf{x}}\|$.

**Assumption 4.** Stochastic gradients $\nabla\ell(\mathbf{x}, \theta)$ are unbiased and variance bounded, i.e., $\mathbb{E}_\theta[\nabla\ell(\mathbf{x}, \theta)] = \nabla L(\mathbf{x})$ and $\mathbb{E}_\theta[\|\nabla\ell(\mathbf{x}, \theta) - \nabla L(\mathbf{x})\|^2] \leq \gamma^2$.

Note the condition in Assumption 4 implies that the local gradients of each node $\nabla f_i(\mathbf{x})$ are also unbiased estimators of the expected risk gradient $\nabla L(\mathbf{x})$ and their variance is bounded above by $\gamma^2/m$ as it is defined as an average over $m$ realizations.

### 4.1 Strongly Convex Setting

This section presents the convergence guarantees of the proposed `QuanTimed-DSGD` method for smooth and strongly convex functions. The following assumption formally defines strong convexity.

**Assumption 5.** The function $\ell$ is $\mu$-strongly convex, i.e., for any $\mathbf{x}, \hat{\mathbf{x}} \in \mathbb{R}^p$ and $\theta \in \mathcal{D}$ we have that $\langle \nabla\ell(\mathbf{x}, \theta) - \nabla\ell(\hat{\mathbf{x}}, \theta), \mathbf{x} - \hat{\mathbf{x}} \rangle \geq \mu\|\mathbf{x} - \hat{\mathbf{x}}\|^2$.

Next, we characterize the convergence rate of `QuanTimed-DSGD` for strongly convex objectives.

**Theorem 1** (Strongly Convex Losses). *If the conditions in Assumptions 1–5 are satisfied and stepsizes are picked as $\alpha = T^{-\delta/2}$ and $\varepsilon = T^{-3\delta/2}$ for arbitrary $\delta \in (0, 1/2)$, then for large enough number of iterations $T \geq T_{\min}^c$ the iterates generated by the `QuanTimed-DSGD` algorithm satisfy*

$$\frac{1}{n}\sum_{i=1}^{n} \mathbb{E}\left[\|\mathbf{x}_{i,T} - \mathbf{x}^*\|^2\right] \leq \mathcal{O}\left(\frac{D^2(K/\mu)^2}{(1-\beta)^2} + \frac{\sigma^2}{\mu}\right)\frac{1}{T^\delta} + \mathcal{O}\left(\frac{\gamma^2}{\mu}\max\left\{\frac{\mathbb{E}[1/V]}{T_d}, \frac{1}{m}\right\}\right)\frac{1}{T^{2\delta}}, \quad (9)$$

*where $D^2 = 2K\sum_{i=1}^{n}(f_i(0) - f_i^*)$, and $f_i^* = \min_{\mathbf{x} \in \mathbb{R}^p} f_i(\mathbf{x})$.*

Theorem 1 guarantees the *exact* convergence of *each* local model to the global optimal even though the noises induced by random quantizations and stochastic gradients are non-vanishing with iterations. Moreover, such convergence rate is as close as desired to $\mathcal{O}(1/\sqrt{T})$ by picking the tuning parameter $\delta$ arbitrarily close to $1/2$. We would like to highlight that by choosing a parameter $\delta$ closer to $1/2$,

the lower bound on the number of required iterations $T^{\mathsf{c}}_{\min}$ becomes larger. More details are available in the proof of Theorem 1 provided in the supplementary material.

Note that the coefficient of $1/T^{\delta}$ in (9) characterizes the dependency of our upper bound on the objective function condition number $K/\mu$, graph connectivity parameter $1/(1-\beta)$, and variance $\sigma^2$ of error induced by quantizing our signals. Moreover, the coefficient of $1/T^{2\delta}$ shows the effect of stochastic gradients variance $\gamma^2$ as well as our deadline-based scheme parameters $T_d/(\mathbb{E}[1/V])$.

**Remark 1.** The expression $1/b_{\mathrm{eff}} = \max\{\mathbb{E}[1/V]/T_d, 1/m\}$ represents the inverse of the effective batch size $b_{\mathrm{eff}}$ used in our `QuanTimed-DSGD` method. To be more specific, If the deadline $T_d$ is large enough that in expectation all local gradients are computed before the deadline, i.e., $T_d/\mathbb{E}[1/V] > m$, then our effective batch size is $b_{\mathrm{eff}} = m$ and the term $1/m$ is the dominant term in the maximization. Conversely, if $T_d$ is small and the number of computed gradients $T_d/\mathbb{E}[1/V]$ is smaller than the total number of local samples $m$, the effective batch size is $b_{\mathrm{eff}} = T_d/\mathbb{E}[1/V]$. In this case, $\mathbb{E}[1/V]/T_d$ is dominant term in the maximization. This observation shows that $\gamma^2 \max\{\mathbb{E}[1/V]/T_d, 1/m\} = \gamma^2/b_{\mathrm{eff}}$ in (9) is the variance of mini-batch gradient in `QuanTimed-DSGD`.

**Remark 2.** Using strong convexity of the objective function, one can easily verify that the last iterates $\mathbf{x}_{i,T}$ of `QuanTimed-DSGD` satisfy the sub-optimality $f(\mathbf{x}_{i,T}) - f(\hat{\mathbf{x}}^*) = L_N(\mathbf{x}_{i,T}) - L_N(\hat{\mathbf{x}}^*) \le \mathcal{O}(1/\sqrt{T})$ with respect to the empirical risk, where $\hat{\mathbf{x}}^*$ is the minimizer of the empirical risk $L_N$. As the gap between the expected risk $L$ and the empirical risk $L_N$ is of $\mathcal{O}(1/\sqrt{mn})$, the overall error of `QuanTimed-DSGD` with respect to the expected risk $L$ is $\mathcal{O}(1/\sqrt{T} + 1/\sqrt{mn})$.

## 4.2 Non-convex Setting

In this section, we characterize the convergence rate of `QuanTimed-DSGD` for non-convex and smooth objectives. As discussed in Section 2, we are interested in finding a set of local models which satisfy first-order optimality condition approximately, while the models are close to each other and satisfy the consensus condition up to a small error. To be more precise, we are interested in finding a set of local models $\{\mathbf{x}_1^*, \dots, \mathbf{x}_n^*\}$ where their average $\overline{\mathbf{x}}^* := \frac{1}{n}\sum_{i=1}^{n} \mathbf{x}_i^*$ (approximately) satisfy first-order optimality condition, i.e., $\mathbb{E}\|\nabla f(\overline{\mathbf{x}}^*)\|^2 \le \nu$, while the iterates are close to their average, i.e., $\mathbb{E}\|\overline{\mathbf{x}}^* - \mathbf{x}_i^*\|^2 \le \rho$. If a set of local iterates satisfies these conditions we call them $(\nu, \rho)$-approximate solutions. Next theorem characterizes both first-order optimality and consensus convergence rates and the overall complexity for achieving an $(\nu, \rho)$-approximate solutions.

**Theorem 2** (Non-convex Losses). *Under Assumptions 1–4, and for step-sizes* $\alpha = T^{-1/6}$ *and* $\varepsilon = T^{-1/2}$, `QuanTimed-DSGD` *guarantees the following convergence and consensus rates:*

$$\frac{1}{T}\sum_{t=0}^{T-1}\mathbb{E}\|\nabla f(\overline{\mathbf{x}}_t)\|^2 \le \mathcal{O}\left(\frac{K^2}{(1-\beta)^2}\frac{\gamma^2}{m} + \frac{K\sigma^2}{n}\right)\frac{1}{T^{1/3}} + \mathcal{O}\left(\frac{K\gamma^2}{n}\max\left\{\frac{\mathbb{E}[1/V]}{T_d}, \frac{1}{m}\right\}\right)\frac{1}{T^{2/3}}, \tag{10}$$

and

$$\frac{1}{T}\sum_{t=0}^{T-1}\frac{1}{n}\sum_{i=1}^{n}\mathbb{E}\|\overline{\mathbf{x}}_t - \mathbf{x}_{i,t}\|^2 \le \mathcal{O}\left(\frac{\gamma^2}{m(1-\beta)^2}\right)\frac{1}{T^{1/3}}, \tag{11}$$

*for large enough number of iterations* $T \ge T^{\mathsf{nc}}_{\min}$*. Here* $\overline{\mathbf{x}}_t = \frac{1}{n}\sum_{i=1}^{n}\mathbf{x}_{i,t}$ *denotes the average models at iteration* $t$*.*

The convergence rate in (10) indicates the proposed `QuanTimed-DSGD` method finds first-order stationary points with vanishing approximation error, even though the quantization and stochastic gradient noises are non-vanishing. Also, the approximation error decays as fast as $\mathcal{O}(T^{-1/3})$ with iterations. Theorem 2 also implies from (11) that the local models reach consensus with a rate of $\mathcal{O}(T^{-1/3})$. Moreover, it shows that to find an $(\nu, \rho)$-approximate solution `QuanTimed-DSGD` requires at most $\mathcal{O}(\max\{\nu^{-3}, \rho^{-3}\})$ iterations.

# 5 Experimental Results

In this section, we numerically evaluate the performance of the proposed `QuanTimed-DSGD` method described in Algorithm 1 for solving a class of non-convex decentralized optimization problems.

In particular, we compare the total run-time of `QuanTimed-DSGD` scheme with the ones for three benchmarks which are briefly described below.

- **Decentralized SGD (DSGD)** (Yuan et al., 2016): Each worker updates its decision variable as $\mathbf{x}_{i,t+1} = \sum_{j \in \mathcal{N}_i} w_{ij} \mathbf{x}_{j,t} - \alpha \widetilde{\nabla} f_i(\mathbf{x}_{i,t})$. We note that the exchanged messages are not quantized and the local gradients are computed for a fixed batch size.

- **Quantized Decentralized SGD (Q-DSGD)** (Reisizadeh et al., 2019a): Iterates are updated according to (8). Similar to `QuanTimed-DSGD` scheme, Q-DSGD employs quantized message-passing, however the gradients are computed for a fixed batch size in each iteration.

- **Asynchronous DSGD**: Each worker updates its model without waiting to receive the updates of its neighbors, i.e. $\mathbf{x}_{i,t+1} = \sum_{j \in \mathcal{N}_i} w_{ij} \mathbf{x}_{j,\tau_j} - \alpha \widetilde{\nabla} f_i(\mathbf{x}_{i,t})$ where $\mathbf{x}_{j,\tau_j}$ denotes the most recent model for node $j$. In our implementation of this scheme, models are exchanged without quantization.

Note that the first two methods mentioned above, i.e., DSGD and Q-DSGD, operate synchronously across the workers, as is our proposed `QuanTimed-DSGD` method. To be more specific, worker nodes wait to receive the decision variables from all of the neighbor nodes and then synchronously update according to an update rule. In `QuanTimed-DSGD` (Figure 1, right), this waiting time consists of a fixed gradient computation time denoted by the deadline $T_d$ and communication time of the message exchanges. Due to the random computation times, different workers end up computing gradients of different and random batch-sizes $B_{i,t}$ across workers $i$ and iterations $t$. In DSGD (and Q-DSGD) however (Figure 1, Left), the gradient computation time varies across the workers since computing a fixed-batch gradient of size $B$ takes a random time whose expected value is proportional to the batch-size $B$ and hence the slowest nodes (stragglers) determine the overall synchronization time $T_{\max}$. Asynchronous-DSGD mitigates stragglers since each worker iteratively computes a gradient of batch-size $B$ and updates the local model using the most recent models of its neighboring nodes available in its memory (Figure 1, middle).

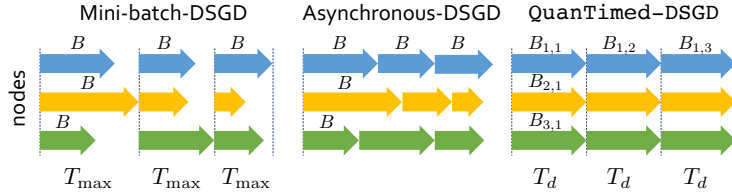

Figure 1: Gradient computation timeline for three methods: DSGD, Asynchronous-DSGD, `QuanTimed-DSGD`.

**Data and Experimental Setup.** We carry out two sets of experiments over CIFAR-10 and MNIST datasets, where each worker is assigned with a sample set of size $m = 200$ for both datasets. For CIFAR-10, we implement a binary classification using a fully connected neural network with one hidden layer with 30 neurons. Each image is converted to a vector of length 1024. For MNIST, we use a fully connected neural network with one hidden layer of size 50 to classify the input image into 10 classes. In experiments over CIFAR-10, step-sizes are fine-tuned as follows: $(\alpha, \varepsilon) = (0.08/T^{1/6}, 14/T^{1/2})$ for `QuanTimed-DSGD` and Q-DSGD, and $\alpha = 0.015$ for DSGD and Asynchronous DSGD. In MNIST experiments, step-sizes are fine-tuned to $(\alpha, \varepsilon) = (0.3/T^{1/6}, 15/T^{1/2})$ for `QuanTimed-DSGD` and Q-DSGD, and $\alpha = 0.2$ for DSGD.

We implement the unbiased low precision quantizer in (7) with various quantization levels $s$, and we let $T_c$ denote the communication time of a $p$-vector without quantization (16-bit precision). The communication time for a quantized vector is then proportioned according the quantization level. In order to ensure that the expected batch size used in each node is a target positive number $b$, we choose the deadline $T_d = b/\mathbb{E}[V]$, where $V \sim \text{Uniform}(10, 90)$ is the random computation speed. The communication graph is a random Erdös-Rènyi graph with edge connectivity $p_c = 0.4$ and $n = 50$ nodes. The weight matrix is designed as $\mathbf{W} = \mathbf{I} - \mathbf{L}/\kappa$ where $\mathbf{L}$ is the Laplacian matrix of the graph and $\kappa > \lambda_{\max}(\mathbf{L})/2$.

**Results.** Figure 2 compares the total training run-time for the `QuanTimed-DSGD` and DSGD schemes. On CIFAR-10 for instance (left), the same (effective) batch-sizes, the proposed `QuanTimed-DSGD` achieves speedups of up to $3\times$ compared to DSGD.

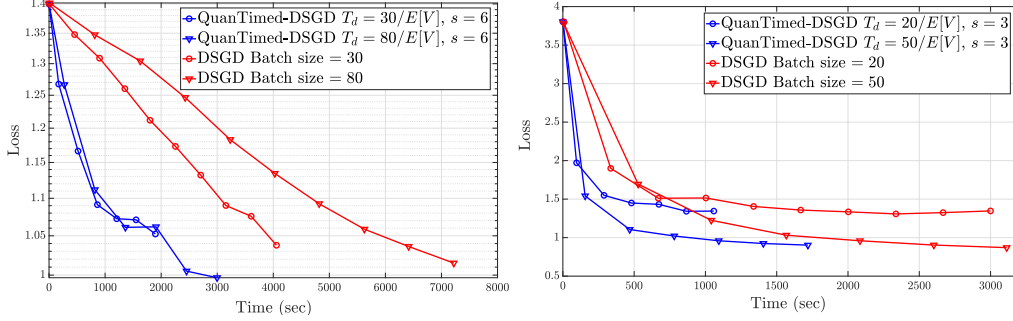

Figure 2: Comparison of `QuanTimed-DSGD` and vanilla DSGD methods for training a neural network on CIFAR-10 (left) and MNIST (right) datasets ($T_c = 3$).

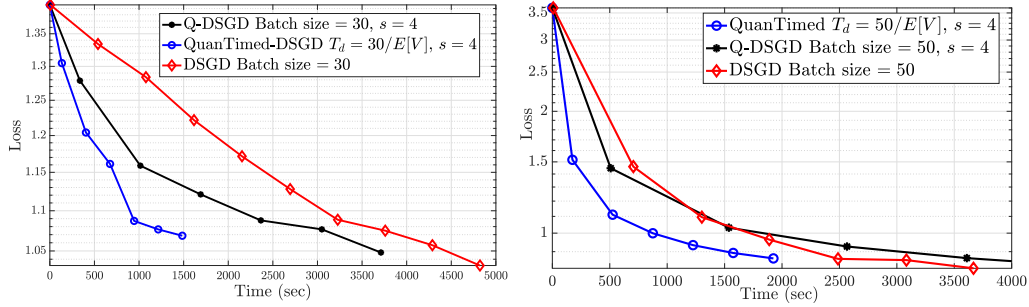

Figure 3: Comparison of `QuanTimed-DSGD`, QDSGD, and vanilla DSGD methods for training a neural network on CIFAR-10 (left) and MNIST (right) datasets ($T_c = 3$).

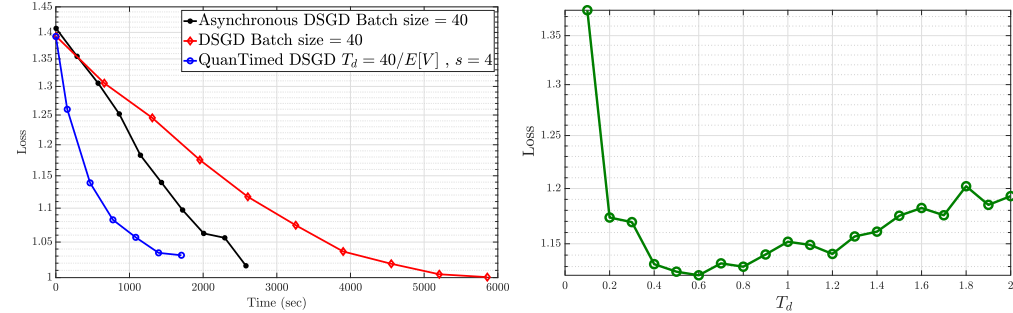

Figure 4: Left: Comparison of `QuanTimed-DSGD` with Asynchronous DSGD and DSGD for training a neural network on CIFAR-10 ($T_c = 3$). Right: Effect of $T_d$ on the loss for CIFAR-10 ($T_c = 1$).

In Figure 3, we further compare these two schemes to Q-DSGD benchmark. Although Q-SGD improves upon the vanilla DSGD by employing quantization, however, the proposed `QuanTimed-DSGD` illustrates $2\times$ speedup in training time over Q-DSGD (left).

To evaluate the straggler mitigation in the `QuanTimed-DSGD`, we compare its run-time with Asynchronous DSGD benchmark in Figure 4 (left). While Asynchronous DSGD outperforms DSGD in training run-time by avoiding slow nodes, the proposed `QuanTimed-DSGD` scheme improves upon Asynchronous DSGD by up to $30\%$. These plots further illustrate that `QuanTimed-DSGD` significantly reduces the training time by simultaneously handling the communication load by quantization and mitigating stragglers through a deadline-based computation. The deadline time $T_d$ indeed can be optimized for the minimum training run-time, as illustrated in Figure 4 (right). Additional numerical results on neural networks with four hidden layers and ImageNet dataset are provided in the supplementary materials.

## 6 Acknowledgments

The authors acknowledge supports from National Science Foundation (NSF) under grant CCF-1909320 and UC Office of President under Grant LFR-18-548175. The research of H. Hassani is supported by NSF grants 1755707 and 1837253.

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
