[Supplementary Material · QuanTimed_DSGD-supplementary.pdf]

# 7 Supplementary Material

## 7.1 Additional Numerical Results

In this section, we provide additional numerical results comparing the performance of the proposed `QuanTimed-DSGD` method with other benchmarks, DSGD and Q-DSGD. In particular, we train a fully connected neural network with four hidden layers consisting of $(30, 20, 20, 25)$ neurons on two classes of CIFAR-10 dataset. For `QuanTimed-DSGD` and Q-DSGD, step sizes are fine-tuned to $(\alpha, \varepsilon) = (0.8/T^{1/6}, 9/T^{1/2})$ and $(\alpha, \varepsilon) = (0.8/T^{1/6}, 10/T^{1/2})$, respectively and for DSGD algorithm $\alpha = 0.08$. Figure 5 (left) demonstrates the training time for the three methods where the proposed `QuanTimed-DSGD` method enjoys a $2.31\times$ speedup over the best of the two benchmarks DSGD and Q-DSGD.

Moreover, we use a one hidden layer neural network with 90 neurons for binary classification on ImageNet dataset and demonstrate the speedup of our proposed method over other benchmarks. For `QuanTimed-DSGD` and Q-DSGD, step sizes are fine-tuned to $(\alpha, \varepsilon) = (0.1/T^{1/6}, 14/T^{1/2})$ and $(\alpha, \varepsilon) = (0.1/T^{1/6}, 15/T^{1/2})$, respectively and for DSGD algorithm $\alpha = 0.02$. Figure 5 (right) shows the training time of the three methods where `QuanTimed-DSGD` demonstrates $1.7\times$ speedup compared to the best of DSGD and Q-DSGD.

Figure 5: Comparison of `QuanTimed-DSGD` with Q-DSGD and DSGD for training a four layer neural network on CIFAR-10 dataset (left) and for training a one layer neural network on ImageNet dataset (right).

## 7.2 Bounding the Stochastic Gradient Noises

In our analysis for both convex and non-convex scenarios, we need to have the noise of various stochastic gradient functions evaluated. Hence, let us start this section by the following lemma which bounds the variance of stochastic gradient functions under our customary Assumption 4.

**Lemma 1.** Assumption 4 results in the followings for any $\mathbf{x} \in \mathbb{R}^p$ and $i \in [n]$:

   (i) $\mathbb{E}[\nabla f_i(\mathbf{x})] = \mathbb{E}[\nabla f(\mathbf{x})] = \nabla L(\mathbf{x})$

   (ii) $\mathbb{E}\left[\left\|\nabla f_i(\mathbf{x}) - \nabla L(\mathbf{x})\right\|^2\right] \leq \frac{\gamma^2}{m}$

   (iii) $\mathbb{E}\left[\left\|\nabla f(\mathbf{x}) - \nabla L(\mathbf{x})\right\|^2\right] \leq \frac{\gamma^2}{nm}$

   (iv) $\mathbb{E}\left[\left\|\nabla f_i(\mathbf{x}) - \nabla f(\mathbf{x})\right\|^2\right] \leq \gamma_1^2 := \gamma^2 \left(\frac{1}{m} + \frac{1}{nm}\right)$

   (v) $\mathbb{E}\left[\widetilde{\nabla} f_i(\mathbf{x})\right] = \nabla L(\mathbf{x})$

   (vi) $\mathbb{E}\left[\left\|\widetilde{\nabla} f_i(\mathbf{x}) - \nabla f_i(\mathbf{x})\right\|^2\right] \leq \gamma_2^2 := 2\gamma^2 \max\left\{\frac{\mathbb{E}[1/V]}{T_d}, \frac{1}{m}\right\}$

*Proof.* The first five expressions (i)-(v) in the lemma are immediate results of Assumption 4 together with the fact that the noise of the stochastic gradient scales down with the sample size. To prove (vi), let $\mathcal{S}_i$ denote the sample set for which node $i$ has computed the gradients. We have

$$
\mathbb{E}\left[\left\|\widetilde{\nabla}f_i(\mathbf{x}) - \nabla L(\mathbf{x})\right\|^2\right] = \sum_b \Pr\left[|\mathcal{S}_i| = b\right]\mathbb{E}\left\|\frac{1}{b}\sum_{\theta \in \mathcal{S}_i}\nabla\ell(\mathbf{x};\theta) - \nabla L(\mathbf{x})\right\|^2
$$

$$
\leq \gamma^2 \sum_b \Pr\left[|\mathcal{S}_i| = b\right]\frac{1}{b}
$$

$$
= \gamma^2 \mathbb{E}[1/|\mathcal{S}_i|]
$$

$$
= \gamma^2 \frac{\mathbb{E}[1/V]}{T_d},
$$

and therefore

$$
\mathbb{E}\left[\left\|\widetilde{\nabla}f_i(\mathbf{x}) - \nabla f_i(\mathbf{x})\right\|^2\right] = \mathbb{E}\left[\left\|\widetilde{\nabla}f_i(\mathbf{x}) - \nabla L(\mathbf{x})\right\|^2\right] + \mathbb{E}\left[\left\|\nabla f_i(\mathbf{x}) - \nabla L(\mathbf{x})\right\|^2\right]
$$

$$
\leq \gamma^2 \left(\frac{\mathbb{E}[1/V]}{T_d} + \frac{1}{m}\right)
$$

$$
\leq 2\gamma^2 \max\left\{\frac{\mathbb{E}[1/V]}{T_d}, \frac{1}{m}\right\}.
$$

$\square$

## 7.3 Proof of Theorem 1

To prove Theorem 1, we first establish two Lemmas 2 and 3 and then easily conclude the theorem from the two results.

The main problem is to minimize the global objective defined in (4). We introduce the following optimization problem which is equivalent the main problem:

$$
\min_{\mathbf{x}_1,\ldots,\mathbf{x}_n \in \mathbb{R}^p} \quad F(\mathbf{x}) := \frac{1}{n}\sum_{i=1}^n f_i(\mathbf{x}_i) \tag{12}
$$

$$
\text{s.t.} \quad \mathbf{x}_i = \mathbf{x}_j, \qquad \text{for all } i, j \in \mathcal{N}_i,
$$

where the vecor $\mathbf{x} = [\mathbf{x}_1; \cdots; \mathbf{x}_n] \in \mathbb{R}^{np}$ denotes the concatenation of all the local models. Clearly, $\widetilde{\mathbf{x}}^* := [\mathbf{x}^*; \cdots; \mathbf{x}^*]$ is the solution to (12). Using Assumption 1, the constraint in the alternative problem (12) can be stated as $(\mathbf{I} - \mathbf{W})^{1/2}\mathbf{x} = 0$. Inspired by this fact, we define the following penalty function for every $\alpha$:

$$
h_\alpha(\mathbf{x}) = \frac{1}{2}\mathbf{x}^\top(\mathbf{I} - \mathbf{W})\mathbf{x} + \alpha n F(\mathbf{x}), \tag{13}
$$

and denote by $\mathbf{x}_\alpha^*$ the (unique) minimizer of $h_\alpha(\mathbf{x})$. That is,

$$
\mathbf{x}_\alpha^* = \arg\min_{\mathbf{x} \in \mathbb{R}^{np}} h_\alpha(\mathbf{x}) = \arg\min_{\mathbf{x} \in \mathbb{R}^{np}} \frac{1}{2}\mathbf{x}^\top(\mathbf{I} - \mathbf{W})\mathbf{x} + \alpha n F(\mathbf{x}). \tag{14}
$$

Next lemma characterizes the deviation of the models generated by the `QuanTimed-DSGD` method at iteration $T$, that is $\mathbf{x}_T = [\mathbf{x}_{1,T}; \cdots; \mathbf{x}_{n,T}]$ from the optimizer of the penalty function, i.e. $\mathbf{x}_\alpha^*$.

**Lemma 2.** Suppose Assumptions 1–5 hold. Then, the expected deviation of the output of `QuanTimed-DSGD` from the solution to Problem (13) is upper bounded by

$$
\mathbb{E}\left[\|\mathbf{x}_T - \mathbf{x}_\alpha^*\|^2\right] \leq \mathcal{O}\left(\frac{n\sigma^2}{\mu}\|W - W_D\|^2\right)\frac{1}{T^\delta} + \mathcal{O}\left(\frac{n\gamma^2}{\mu}\left(\frac{\mathbb{E}[1/V]}{T_d} + \frac{1}{m}\right)\right)\frac{1}{T^{2\delta}}, \tag{15}
$$

for $\varepsilon = T^{-3\delta/2}$, $\alpha = T^{-\delta/2}$, any $\delta \in (0, 1/2)$ and $T \geq T^c_{\text{min1}}$, where

$$
T^c_{\text{min1}} := \max\left\{\left\lceil\left(\frac{(2+K)^2}{\mu}\right)^{\frac{1}{\delta}}\right\rceil, \left\lceil e^{e^{\frac{1}{1-2\delta}}}\right\rceil, \left\lceil\mu^{\frac{1}{2\delta}}\right\rceil\right\}. \tag{16}
$$

*Proof of Lemma 2.* First note that the gradient of the penalty function $h_\alpha$ defined in (13) is as follows:

$$\nabla h_\alpha(\mathbf{x}_t) = (\mathbf{I} - \mathbf{W})\mathbf{x}_t + \alpha n \nabla F(\mathbf{x}_t), \tag{17}$$

where $\mathbf{x}_t = [\mathbf{x}_{1,t}; \cdots; \mathbf{x}_{n,t}]$ denotes the concatenation of models at iteration $t$. Now consider the following stochastic gradient function for $h_\alpha$:

$$\widetilde{\nabla} h_\alpha(\mathbf{x}_t) = (\mathbf{W}_D - \mathbf{W})\mathbf{z}_t + (\mathbf{I} - \mathbf{W}_D)\mathbf{x}_t + \alpha n \widetilde{\nabla} F(\mathbf{x}_t), \tag{18}$$

where

$$\widetilde{\nabla} F(\mathbf{x}_t) = \left[ \frac{1}{n} \widetilde{\nabla} f_1(\mathbf{x}_{1,t}); \cdots; \frac{1}{n} \widetilde{\nabla} f_n(\mathbf{x}_{n,t}) \right].$$

We let $\mathcal{F}^t$ denote a sigma algebra that measures the history of the system up until time $t$. According to Assumptions 2 and 4, the stochastic gradient defined above is unbiased, that is,

$$\begin{aligned}
\mathbb{E}\left[ \widetilde{\nabla} h_\alpha(\mathbf{x}_t) | \mathcal{F}^t \right] &= (\mathbf{W}_D - \mathbf{W}) \mathbb{E}\left[ \mathbf{z}_t | \mathcal{F}^t \right] + (\mathbf{I} - \mathbf{W}_D)\mathbf{x}_t + \alpha n \mathbb{E}\left[ \widetilde{\nabla} F(\mathbf{x}_t) | \mathcal{F}^t \right] \\
&= (\mathbf{I} - \mathbf{W})\mathbf{x}_t + \alpha n \nabla F(\mathbf{x}_t) \\
&= \nabla h_\alpha(\mathbf{x}_t).
\end{aligned}$$

We can also write the update rule of `QuanTimed-DSGD` method as follows:

$$\begin{aligned}
\mathbf{x}_{t+1} &= \mathbf{x}_t - \varepsilon \left( (\mathbf{W}_D - \mathbf{W})\mathbf{z}_t + (\mathbf{I} - \mathbf{W}_D)\mathbf{x}_t + \alpha n \widetilde{\nabla} F(\mathbf{x}_t) \right) \\
&= \mathbf{x}_t - \varepsilon \widetilde{\nabla} h_\alpha(\mathbf{x}_t), \tag{19}
\end{aligned}$$

which also represents an iteration of the Stochastic Gradient Descent (SGD) algorithm with step-size $\varepsilon$ in order to minimize the penalty function $h_\alpha(\mathbf{x})$ over $\mathbf{x} \in \mathbb{R}^{np}$. We can bound the deviation of the iteration generated by `QuanTimed-DSGD` from the optimizer $\mathbf{x}_\alpha^*$ as follows:

$$\begin{aligned}
\mathbb{E}\left[ \|\mathbf{x}_{t+1} - \mathbf{x}_\alpha^*\|^2 | \mathcal{F}^t \right] &= \mathbb{E}\left[ \left\| \mathbf{x}_t - \varepsilon \widetilde{\nabla} h_\alpha(\mathbf{x}_t) - \mathbf{x}_\alpha^* \right\|^2 | \mathcal{F}^t \right] \\
&= \|\mathbf{x}_t - \mathbf{x}_\alpha^*\|^2 - 2\varepsilon \left\langle \mathbf{x}_t - \mathbf{x}_\alpha^*, \mathbb{E}\left[ \widetilde{\nabla} h_\alpha(\mathbf{x}_t) | \mathcal{F}^t \right] \right\rangle \\
&\quad + \varepsilon^2 \mathbb{E}\left[ \left\| \widetilde{\nabla} h_\alpha(\mathbf{x}_t) \right\|^2 | \mathcal{F}^t \right] \\
&= \|\mathbf{x}_t - \mathbf{x}_\alpha^*\|^2 - 2\varepsilon \left\langle \mathbf{x}_t - \mathbf{x}_\alpha^*, \nabla h_\alpha(\mathbf{x}_t) \right\rangle \\
&\quad + \varepsilon^2 \mathbb{E}\left[ \left\| \widetilde{\nabla} h_\alpha(\mathbf{x}_t) \right\|^2 | \mathcal{F}^t \right] \\
&\le (1 - 2\mu_\alpha \varepsilon) \|\mathbf{x}_t - \mathbf{x}_\alpha^*\|^2 + \varepsilon^2 \mathbb{E}\left[ \left\| \widetilde{\nabla} h_\alpha(\mathbf{x}_t) \right\|^2 | \mathcal{F}^t \right], \tag{20}
\end{aligned}$$

where we used the fact that the penalty function $h_\alpha$ is strongly convex with parameter $\mu_\alpha := \alpha \mu$. Moreover, we can bound the second term in RHS of (20) as follows:

$$\begin{aligned}
&\mathbb{E}\left[ \left\| \widetilde{\nabla} h_\alpha(\mathbf{x}_t) \right\|^2 | \mathcal{F}^t \right] \\
&= \mathbb{E}\left[ \left\| (\mathbf{W}_D - \mathbf{W})\mathbf{z}_t + (\mathbf{I} - \mathbf{W}_D)\mathbf{x}_t + \alpha n \widetilde{\nabla} F(\mathbf{x}_t) \right\|^2 | \mathcal{F}^t \right] \\
&= \mathbb{E}\left[ \left\| (\mathbf{I} - \mathbf{W})\mathbf{x}_t + \alpha n \nabla F(\mathbf{x}_t) + (\mathbf{W}_D - \mathbf{W})(\mathbf{z}_t - \mathbf{x}_t) + \alpha n \widetilde{\nabla} F(\mathbf{x}_t) - \alpha n \nabla F(\mathbf{x}_t) \right\|^2 | \mathcal{F}^t \right] \\
&= \left\| \nabla h_\alpha(\mathbf{x}_t) \right\|^2 + \mathbb{E}\left[ \left\| (\mathbf{W}_D - \mathbf{W})(\mathbf{z}_t - \mathbf{x}_t) \right\|^2 | \mathcal{F}^t \right] + \alpha^2 n^2 \mathbb{E}\left[ \left\| \widetilde{\nabla} F(\mathbf{x}_t) - \nabla F(\mathbf{x}_t) \right\|^2 | \mathcal{F}^t \right] \\
&\le K_\alpha^2 \|\mathbf{x}_t - \mathbf{x}_\alpha^*\|^2 + n\sigma^2 \|W - W_D\|^2 + \alpha^2 n \gamma_2^2. \tag{21}
\end{aligned}$$

To derive (21), we used the facts that $h_\alpha$ is smooth with parameter $K_\alpha := 1 - \lambda_n(W) + \alpha K$; the quantizer is unbiased with variance $\leq \sigma^2$ (Assumption 2); stochastic gradients of the loss function are unbiased and variance-bounded (Assumption 4 and Lemma 1). Plugging (21) in (20) yields

$$\mathbb{E}\left[\|\mathbf{x}_{t+1} - \mathbf{x}_\alpha^*\|^2 \,|\mathcal{F}^t\right] \leq \left(1 - 2\mu_\alpha\varepsilon + \varepsilon^2 K_\alpha^2\right)\|\mathbf{x}_t - \mathbf{x}_\alpha^*\|^2 + \varepsilon^2 n\sigma^2\|W - W_D\|^2 + \alpha^2\varepsilon^2 n\gamma_2^2. \tag{22}$$

To ease the notation, let $e_t := \mathbb{E}[\|\mathbf{x}_t - \mathbf{x}_\alpha^*\|^2]$ denote the expected deviation of the models at iteration $t$ i.e. $\mathbf{x}_t$ from the optimizer $\mathbf{x}_\alpha^*$ with respect to all the randomnesses from iteration $t = 0$. Therefore,

$$e_{t+1} \leq \left(1 - 2\mu_\alpha\varepsilon + \varepsilon^2 K_\alpha^2\right) e_t + \varepsilon^2 n\sigma^2\|W - W_D\|^2 + \alpha^2\varepsilon^2\gamma_2^2$$
$$= \left(1 - \varepsilon(2\mu_\alpha - \varepsilon K_\alpha^2)\right) e_t + \varepsilon^2 n\sigma^2\|W - W_D\|^2 + \alpha^2\varepsilon^2 n\gamma_2^2. \tag{23}$$

For any $T \geq T_{\mathsf{min1}}^{\mathsf{c}}$ and the proposed pick $\varepsilon = T^{-3\delta/2}$, we have

$$T^\delta \geq (T_{\mathsf{min1}}^{\mathsf{c}})^\delta \geq \frac{(2 + K)^2}{\mu},$$

and therefore

$$\varepsilon = \frac{1}{T^{3\delta/2}}$$
$$\leq \frac{\mu}{(2 + K)^2} \cdot \frac{1}{T^{\delta/2}}$$
$$\leq \frac{\mu_\alpha}{(2 + \alpha K)^2}$$
$$\leq \frac{\mu_\alpha}{K_\alpha^2}.$$

Hence, we can further bound (23) as follows:

$$e_{t+1} \leq \left(1 - \varepsilon\left(2\mu_\alpha - \varepsilon K_\alpha^2\right)\right) e_t + \varepsilon^2 n\sigma^2\|W - W_D\|^2 + \alpha^2\varepsilon^2 n\gamma_2^2$$
$$\leq (1 - \mu_\alpha\varepsilon) e_t + \varepsilon^2 n\sigma^2\|W - W_D\|^2 + \alpha^2\varepsilon^2 n\gamma_2^2$$
$$= \left(1 - \frac{\mu}{T^{2\delta}}\right) e_t + \frac{n\sigma^2\|W - W_D\|^2}{T^{3\delta}} + \frac{n\gamma_2^2}{T^{4\delta}}.$$

Now, we let $(a, b, c) = (\mu, n\sigma^2\|W - W_D\|^2, n\gamma_2^2)$ and employ Lemma 4 which yields

$$e_T = \mathbb{E}\left[\|\mathbf{x}_T - \mathbf{x}_\alpha^*\|^2\right]$$
$$\leq \mathcal{O}\left(\frac{b/a}{T^\delta}\right) + \mathcal{O}\left(\frac{c/a}{T^{2\delta}}\right)$$
$$= \mathcal{O}\left(\frac{n\sigma^2}{\mu}\|W - W_D\|^2 \frac{1}{T^\delta}\right) + \mathcal{O}\left(\frac{n\gamma^2}{\mu}\left(\frac{\mathbb{E}[1/V]}{T_d} + \frac{1}{m}\right)\frac{1}{T^{2\delta}}\right),$$

and the proof of Lemma 2 is concluded. $\square$

Now we also bound the deviation of the optimizers of the penalty function and the main loss function, that is $\mathbf{x}_\alpha^*$ and $\widetilde{\mathbf{x}}^*$.

**Lemma 3.** Suppose Assumptions 1, 3–5 hold. Then the difference between the optimal solutions to (12) and its penalized version (14) is bounded above by

$$\|\mathbf{x}_\alpha^* - \widetilde{\mathbf{x}}^*\| \leq \mathcal{O}\left(\frac{\sqrt{2n}c_2 D\left(3 + 2K/\mu\right)}{1 - \beta}\frac{1}{T^{\delta/2}}\right),$$

for $\alpha = T^{-\delta/2}$, any $\delta \in (0, 1/2)$ and $T \geq T_{\mathsf{min2}}^{\mathsf{c}}$ where

$$T_{\mathsf{min2}}^{\mathsf{c}} := \max\left\{\left\lceil\left(\frac{K}{1 + \lambda_n(W)}\right)^{\frac{2}{\delta}}\right\rceil, \left\lceil(\mu + K)^{\frac{2}{\delta}}\right\rceil\right\}.$$

*Proof of Lemma 3.* First, recall the penalty function minimization in (14). Following sequence is the update rule associated with this problem when the gradient descent method is applied to the objective function $h_\alpha$ with the unit step-size $\eta = 1$,

$$\mathbf{u}_{t+1} = \mathbf{u}_t - \eta \nabla h_\alpha(\mathbf{u}_t) = \mathbf{W}\mathbf{u}_t - \alpha n \nabla F(\mathbf{u}_t). \tag{24}$$

From analysis of GD for strongly convex objectives, the sequence $\{\mathbf{u}_t : t = 0, 1, \cdots\}$ defined above exponentially converges to the minimizer of $h_\alpha$, $\mathbf{x}_\alpha^*$, provided that $1 = \eta \leq 2/K_\alpha$. The latter condition is satisfied if we make $\alpha \leq (1 + \lambda_n(W))/K$. Therefore,

$$\|\mathbf{u}_t - \mathbf{x}_\alpha^*\|^2 \leq (1 - \mu_\alpha)^t \|\mathbf{u}_0 - \mathbf{x}_\alpha^*\|^2$$
$$= (1 - \alpha\mu)^t \|\mathbf{u}_0 - \mathbf{x}_\alpha^*\|^2.$$

If we take $\mathbf{u}_0 = 0$, then (24) implies

$$\|\mathbf{u}_T - \mathbf{x}_\alpha^*\|^2 \leq (1 - \alpha\mu)^T \|\mathbf{x}_\alpha^*\|^2$$
$$\leq 2(1 - \alpha\mu)^T \left( \|\widetilde{\mathbf{x}}^* - \mathbf{x}_\alpha^*\|^2 + \|\widetilde{\mathbf{x}}^*\|^2 \right)$$
$$= 2(1 - \alpha\mu)^T \left( \|\widetilde{\mathbf{x}}^* - \mathbf{x}_\alpha^*\|^2 + n\|\mathbf{x}^*\|^2 \right), \tag{25}$$

On the other hand, it can be shown (Yuan et al. (2016)) that if $\alpha \leq \min\{(1+\lambda_n(W))/K, 1/(\mu+K)\}$, then the sequence $\{\mathbf{u}_t : t = 0, 1, \cdots\}$ defined in (24) converges to the $\mathcal{O}(\frac{\alpha}{1-\beta})$-neighborhood of the optima $\widetilde{\mathbf{x}}^*$, i.e.,

$$\|\mathbf{u}_t - \widetilde{\mathbf{x}}^*\| \leq \mathcal{O}\left( \frac{\alpha}{1-\beta} \right). \tag{26}$$

If we take $\alpha = T^{-\delta/2}$, the condition $T \geq T_{\mathsf{min\text{-}c2}}$ implies that $\alpha \leq \min\{(1+\lambda_n(W))/K, 1/(\mu+K)\}$. Therefore, (26) yields

$$\|\mathbf{u}_T - \widetilde{\mathbf{x}}^*\| \leq \mathcal{O}\left( \frac{\alpha}{1-\beta} \right). \tag{27}$$

More precisely, we have the following (See Corollary 9 in Yuan et al. (2016)):

$$\|\mathbf{u}_T - \widetilde{\mathbf{x}}^*\| \leq \sqrt{n} \left( c_3^T \|\mathbf{x}^*\| + \frac{c_4}{\sqrt{1-c_3^2}} + \frac{\alpha D}{1-\beta} \right), \tag{28}$$

where

$$c_3^2 = 1 - \frac{1}{2} \cdot \frac{\mu K}{\mu + K} \alpha,$$

$$\frac{c_4}{\sqrt{1-c_3^2}} = \frac{\alpha K D}{1-\beta} \sqrt{4 \left( \frac{\mu+K}{\mu K} \right)^2 - 2 \cdot \frac{\mu+K}{\mu K} \alpha}$$
$$\leq \frac{2\alpha D}{(1-\beta)} \left( 1 + K/\mu \right).$$

$$D^2 = 2K \sum_{i=1}^{n} \left( f_i(0) - f_i^* \right), \quad f_i^* = \min_{\mathbf{x} \in \mathbb{R}^p} f_i(\mathbf{x}).$$

From (28) and (27), we have for $T \geq T_2$

$$\|\mathbf{x}_\alpha^* - \widetilde{\mathbf{x}}^*\|^2 = \|\mathbf{x}_\alpha^* - \mathbf{u}_T + \mathbf{u}_T - \widetilde{\mathbf{x}}^*\|^2$$
$$\leq 2\|\mathbf{x}_\alpha^* - \mathbf{u}_T\|^2 + 2\|\mathbf{u}_T - \widetilde{\mathbf{x}}^*\|^2$$
$$\leq 4(1-\alpha\mu)^T \left( \|\widetilde{\mathbf{x}}^* - \mathbf{x}_\alpha^*\|^2 + n\|\mathbf{x}^*\|^2 \right)$$
$$+ 2n \left( \left( 1 - \frac{1}{2} \cdot \frac{\mu K}{\mu + K} \alpha \right)^{T/2} \|\mathbf{x}^*\| + \frac{\alpha D}{1-\beta} \left( 3 + 2K/\mu \right) \right)^2. \tag{29}$$

Note that for our pick $\alpha = T^{-\delta/2}$, we can write

$$(1 - \alpha\mu)^T \le \exp\left(-T^{1-\delta/2}\right) =: e_1(T),$$

$$\left(1 - \frac{1}{2} \cdot \frac{\mu K}{\mu + K}\alpha\right)^{T/2} \le \exp\left(-\frac{1}{2} \cdot \frac{\mu K}{\mu + K}T^{1-\delta/2}\right) =: e_2(T).$$

Therefore, from (29) we have

$$\|\mathbf{x}_\alpha^* - \widetilde{\mathbf{x}}^*\|^2$$

$$\le \frac{1}{(1 - 4e_1(T))}\left\{4e_1(T)n\|\mathbf{x}^*\|^2 + 2ne_2^2(T)\|\mathbf{x}^*\|^2 + 4ne_2(T)\|\mathbf{x}^*\|\frac{\alpha D}{1 - \beta}\left(3 + 2K/\mu\right)\right.$$

$$\left. + 2nD^2\left(3 + 2K/\mu\right)^2\left(\frac{\alpha}{1-\beta}\right)^2\right\}$$

$$\le \frac{4n\left(2e_1(T) + e_2^2(T)\right)}{(1 - 4e_1(T))}\frac{f_0 - f^*}{\mu} + \frac{4\sqrt{2}ne_2(T)}{(1 - 4e_1(T))}\sqrt{\frac{f_0 - f^*}{\mu}}\frac{\alpha D}{1 - \beta}\left(3 + 2K/\mu\right)$$

$$+ \frac{2nD^2\left(3 + 2K/\mu\right)^2}{(1 - 4e_1(T))}\left(\frac{\alpha}{1-\beta}\right)^2, \tag{30}$$

where we used the fact that $\|\mathbf{x}^*\|^2 \le 2(f_0 - f^*)/\mu$ for $f_0 = f(0)$ and $f^* = \min_{\mathbf{x}\in\mathbb{R}^p} f(\mathbf{x}) = f(\mathbf{x}^*)$. Given the fact that the terms $e_1(T)$ and $e_2(T)$ decay exponentially, i.e. $e_1(T) = o\left(\alpha^2\right)$ and $e_2(T) = o\left(\alpha^2\right)$, we have

$$\|\mathbf{x}_\alpha^* - \widetilde{\mathbf{x}}^*\| \le \mathcal{O}\left(\sqrt{2n}D\left(3 + 2K/\mu\right)\frac{\alpha}{1 - \beta}\right)$$

$$= \mathcal{O}\left(\frac{\sqrt{2n}D\left(3 + 2K/\mu\right)}{1 - \beta}\frac{1}{T^{\delta/2}}\right),$$

which concludes the claim in Lemma 3. □

Having proved Lemmas 2 and 3, we can now plug them in Theorem 1 and write for $T \ge T_{\min}^{\mathsf{c}} := \max\{T_{\min1}^{\mathsf{c}}, T_{\min2}^{\mathsf{c}}\}$

$$\frac{1}{n}\sum_{i=1}^n \mathbb{E}\left[\|\mathbf{x}_{i,T} - \mathbf{x}^*\|^2\right] = \frac{1}{n}\mathbb{E}\left[\|\mathbf{x}_T - \widetilde{\mathbf{x}}^*\|^2\right]$$

$$= \frac{1}{n}\mathbb{E}\left[\|\mathbf{x}_T - \mathbf{x}_\alpha^* + \mathbf{x}_\alpha^* - \widetilde{\mathbf{x}}^*\|^2\right]$$

$$\le \frac{2}{n}\mathbb{E}\left[\|\mathbf{x}_T - \mathbf{x}_\alpha^*\|^2\right] + \frac{2}{n}\|\mathbf{x}_\alpha^* - \widetilde{\mathbf{x}}^*\|^2$$

$$\le \mathcal{O}\left(\frac{D^2(K/\mu)^2}{(1-\beta)^2} + \frac{\sigma^2}{\mu}\right)\frac{1}{T^\delta} + \mathcal{O}\left(\frac{\gamma^2}{\mu}\max\left\{\frac{\mathbb{E}[1/V]}{T_d}, \frac{1}{m}\right\}\right)\frac{1}{T^{2\delta}}.$$

In the end, we state and proof Lemma 4 which we used its result earlier in the proof of Lemma 2.

**Lemma 4.** Let the non-negative sequence $e_t$ satisfy the inequality

$$e_{t+1} \le \left(1 - \frac{a}{T^{2\delta}}\right)e_t + \frac{b}{T^{3\delta}} + \frac{c}{T^{3\delta}}, \tag{31}$$

for $t = 0, 1, 2, \cdots$, positive constants $a, b, c$ and $\delta \in (0, 1/2)$. Then, after

$$T \ge \max\left\{\left\lceil e^{e^{\frac{1}{1-2\delta}}}\right\rceil, \left\lceil a^{\frac{1}{2\delta}}\right\rceil\right\}$$

iterations, the iterate $e_T$ satisfies

$$e_T \le \mathcal{O}\left(\frac{b/a}{T^\delta}\right) + \mathcal{O}\left(\frac{c/a}{T^{2\delta}}\right). \tag{32}$$

*Proof of Lemma 4.* Use the expression in (31) for steps $t-1$ and $t$ to obtain

$$e_{t+1} \le \left(1 - \frac{a}{T^{2\delta}}\right)^2 e_{t-1} + \left[1 + \left(1 - \frac{a}{T^{2\delta}}\right)\right] \frac{b}{T^{3\delta}} + \left[1 + \left(1 - \frac{a}{T^{2\delta}}\right)\right] \frac{c}{T^{4\delta}},$$

where $T \ge a^{1/(2\delta)}$. By recursively applying these inequalities for all steps $t = 0, 1, \cdots$ we obtain that

$$
\begin{aligned}
e_t &\le \left(1 - \frac{a}{T^{2\delta}}\right)^t e_0 \\
&\quad + \frac{b}{T^{3\delta}} \left[1 + \left(1 - \frac{a}{T^{2\delta}}\right) + \cdots + \left(1 - \frac{a}{T^{2\delta}}\right)^{t-1}\right] \\
&\quad + \frac{c}{T^{4\delta}} \left[1 + \left(1 - \frac{a}{T^{2\delta}}\right) + \cdots + \left(1 - \frac{a}{T^{2\delta}}\right)^{t-1}\right] \\
&\le \left(1 - \frac{a}{T^{2\delta}}\right)^t e_0 + \frac{b}{T^{3\delta}} \left[\sum_{s=0}^{t-1} \left(1 - \frac{a}{T^{2\delta}}\right)^s\right] + \frac{c}{T^{4\delta}} \left[\sum_{s=0}^{t-1} \left(1 - \frac{a}{T^{2\delta}}\right)^s\right] \\
&\le \left(1 - \frac{a}{T^{2\delta}}\right)^t e_0 + \frac{b}{T^{3\delta}} \left[\sum_{s=0}^{\infty} \left(1 - \frac{a}{T^{2\delta}}\right)^s\right] + \frac{c}{T^{4\delta}} \left[\sum_{s=0}^{\infty} \left(1 - \frac{a}{T^{2\delta}}\right)^s\right] \\
&= \left(1 - \frac{a}{T^{2\delta}}\right)^t e_0 + \frac{b}{T^{3\delta}} \left[\frac{1}{1 - \left(1 - \frac{a}{T^{2\delta}}\right)}\right] + \frac{c}{T^{4\delta}} \left[\frac{1}{1 - \left(1 - \frac{a}{T^{2\delta}}\right)}\right] \\
&= \left(1 - \frac{a}{T^{2\delta}}\right)^t e_0 + \frac{b/a}{T^{\delta}} + \frac{c/a}{T^{2\delta}}.
\end{aligned}
$$

Therefore, for the iterate corresponding to step $t = T$ we can write

$$
\begin{aligned}
e_T &\le \left(1 - \frac{a}{T^{2\delta}}\right)^T e_0 + \frac{b/a}{T^{\delta}} + \frac{c/a}{T^{2\delta}} \\
&\le \exp\left(-aT^{(1-2\delta)}\right) e_0 + \frac{b/a}{T^{\delta}} + \frac{c/a}{T^{2\delta}} \\
&= \mathcal{O}\left(\frac{b/a}{T^{\delta}}\right) + \mathcal{O}\left(\frac{c/a}{T^{2\delta}}\right),
\end{aligned}
\tag{33}
$$

and the claim in (32) follows. Note that for the last inequality we assumed that the exponential term in is negligible comparing to the sublinear term. It can be verified for instance if $1 - 2\delta$ is of $\mathcal{O}\left(1/\log(\log(T))\right)$ or greater than that, it satisfies this condition. Moreover, setting $\delta = 1/2$ results in a constant (and hence non-vanishing) term in (33). □

## 7.4 Proof of Theorem 2

To ease the notation, we agree in this section on the following shorthand notations for $t = 0, 1, 2, \cdots$:

$$
\begin{aligned}
X_t &= \begin{bmatrix} \mathbf{x}_{1,t} & \cdots & \mathbf{x}_{n,t} \end{bmatrix} \in \mathbb{R}^{p \times n}, \\
Z_t &= \begin{bmatrix} \mathbf{z}_{1,t} & \cdots & \mathbf{z}_{n,t} \end{bmatrix} \in \mathbb{R}^{p \times n}, \\
\bar{\mathbf{x}}_t &= \frac{1}{n} \sum_{i=1}^{n} \mathbf{x}_{i,t} \in \mathbb{R}^p, \\
\bar{X}_t &= \begin{bmatrix} \bar{\mathbf{x}}_t & \cdots & \bar{\mathbf{x}}_t \end{bmatrix} \in \mathbb{R}^{p \times n}, \\
\widetilde{\partial} f(X_t) &= \begin{bmatrix} \widetilde{\nabla} f_1(\mathbf{x}_{1,t}) & \cdots & \widetilde{\nabla} f_n(\mathbf{x}_{n,t}) \end{bmatrix} \in \mathbb{R}^{p \times n}, \\
\partial f(X_t) &= \begin{bmatrix} \nabla f_1(\mathbf{x}_{1,t}) & \cdots & \nabla f_n(\mathbf{x}_{n,t}) \end{bmatrix} \in \mathbb{R}^{p \times n}.
\end{aligned}
$$

As stated before, we can write the update rule of the proposed `QuanTimed-DSGD` in the following matrix form:

$$X_{t+1} = X_t \left((1-\varepsilon)I + \varepsilon W\right) + \varepsilon(Z_t - X_t)(W - W_D) - \alpha\varepsilon\widetilde{\partial}f(X_t). \tag{34}$$

Let us denote $W_\varepsilon = (1-\varepsilon)I + \varepsilon W$ and write (34) as

$$X_{t+1} = X_t W_\varepsilon + \varepsilon(Z_t - X_t)(W - W_D) - \alpha\varepsilon\widetilde{\partial}f(X_t). \tag{35}$$

Clearly for any $\varepsilon \in (0, 1]$, $W_\varepsilon$ is also doubly stochastic with eigenvalues $\lambda_i(W_\varepsilon) = 1 - \varepsilon + \varepsilon\lambda_i(W)$ and spectral gap $1 - \beta_\varepsilon = 1 - \max\left\{|\lambda_2(W_\varepsilon)|, |\lambda_n(W_\varepsilon)|\right\}$.

We start the convergence analysis by using the smoothness property of the objectives and write

$$
\begin{aligned}
\mathbb{E}f\left(\frac{X_{t+1}\mathbf{1}_n}{n}\right) &= \mathbb{E}f\left(\frac{X_t W_\varepsilon \mathbf{1}_n}{n} + \frac{\varepsilon(Z_t - X_t)(W - W_D)\mathbf{1}_n}{n} - \frac{\alpha\varepsilon\widetilde{\partial}f(X_t)\mathbf{1}_n}{n}\right) \\
&\overset{\text{Assumption } 3}{\leq} \mathbb{E}f\left(\frac{X_t \mathbf{1}_n}{n}\right) - \alpha\varepsilon\mathbb{E}\left\langle\nabla f\left(\frac{X_t\mathbf{1}_n}{n}\right), \frac{\partial f(X_t)\mathbf{1}_n}{n}\right\rangle \\
&\quad + \frac{\varepsilon^2 K}{2}\mathbb{E}\left\|\frac{(Z_t - X_t)(W - W_D)\mathbf{1}_n}{n} - \alpha\frac{\widetilde{\partial}f(X_t)\mathbf{1}_n}{n}\right\|^2. 
\end{aligned}\tag{36}
$$

We specifically used the following equivalent form of the smoothness (Assumption 3) for every local and hence the global objective

$$f_i(\mathbf{y}) \leq f_i(\mathbf{x}) + \left\langle\nabla f_i(\mathbf{x}), \mathbf{y} - \mathbf{x}\right\rangle + \frac{K}{2}\|\mathbf{y} - \mathbf{x}\|^2, \quad \text{for all } i \in [n], \mathbf{x}, \mathbf{y} \in \mathbb{R}^p.$$

Also, we used the following simple fact:

$$W_\varepsilon\mathbf{1}_n = ((1-\varepsilon)I + \varepsilon W)\mathbf{1}_n = (1-\varepsilon)\mathbf{1}_n + \varepsilon W\mathbf{1}_n = \mathbf{1}_n$$

Now let us bound the term in (36) as follows:

$$
\begin{aligned}
\mathbb{E}\left\|\frac{(Z_t - X_t)(W - W_D)\mathbf{1}_n}{n} - \alpha\frac{\widetilde{\partial}f(X_t)\mathbf{1}_n}{n}\right\|^2 &= \mathbb{E}\left\|\frac{(Z_t - X_t)(W - W_D)\mathbf{1}_n}{n}\right\|^2 \\
&\quad + \mathbb{E}\left\|\alpha\frac{\widetilde{\partial}f(X_t)\mathbf{1}_n}{n}\right\|^2 \\
&= \frac{1}{n^2}\sum_{i=1}^{n}(1 - w_{ii})^2\mathbb{E}\left\|\mathbf{z}_{i,t} - \mathbf{x}_{i,t}\right\|^2 \\
&\quad + \alpha^2\mathbb{E}\left\|\frac{\widetilde{\partial}f(X_t)\mathbf{1}_n}{n}\right\|^2 \\
&\leq \frac{\sigma^2}{n} + \alpha^2\mathbb{E}\left\|\frac{\widetilde{\partial}f(X_t)\mathbf{1}_n}{n}\right\|^2, 
\end{aligned}\tag{37}
$$

where we used Assumption 2 to derive the first term in (37). To bound the second term in (37), we have

$$
\mathbb{E}\left\|\frac{\widetilde{\partial} f(X_t)\mathbf{1}_n}{n}\right\|^2 = \mathbb{E}\left\|\frac{\sum_{i=1}^n \widetilde{\nabla} f_i(\mathbf{x}_{i,t})}{n}\right\|^2
$$

$$
= \mathbb{E}\left\|\frac{\sum_{i=1}^n \widetilde{\nabla} f_i(\mathbf{x}_{i,t}) - \nabla f_i(\mathbf{x}_{i,t}) + \nabla f_i(\mathbf{x}_{i,t})}{n}\right\|^2
$$

$$
= \mathbb{E}\left\|\frac{\sum_{i=1}^n \widetilde{\nabla} f_i(\mathbf{x}_{i,t}) - \nabla f_i(\mathbf{x}_{i,t})}{n}\right\|^2 + \mathbb{E}\left\|\frac{\sum_{i=1}^n \nabla f_i(\mathbf{x}_{i,t})}{n}\right\|^2
$$

$$
\leq \frac{\gamma^2}{n}\left(\frac{\mathbb{E}[1/V]}{T_d} + \frac{1}{m}\right) + \mathbb{E}\left\|\frac{\sum_{i=1}^n \nabla f_i(\mathbf{x}_{i,t})}{n}\right\|^2
$$

$$
= \frac{\gamma_2^2}{n} + \mathbb{E}\left\|\frac{\sum_{i=1}^n \nabla f_i(\mathbf{x}_{i,t})}{n}\right\|^2. \tag{38}
$$

where the last inequality follows from Lemma 1.

Plugging (38) in (36) yields

$$
\mathbb{E}f\left(\frac{X_{t+1}\mathbf{1}_n}{n}\right) \leq \mathbb{E}f\left(\frac{X_t\mathbf{1}_n}{n}\right) - \alpha\varepsilon \mathbb{E}\left\langle \nabla f\left(\frac{X_t\mathbf{1}_n}{n}\right), \frac{\partial f(X_t)\mathbf{1}_n}{n}\right\rangle
$$

$$
+ \frac{\varepsilon^2 K}{2n}\sigma^2 + \frac{\alpha^2\varepsilon^2 K}{2n}\gamma_2^2 + \frac{\alpha^2\varepsilon^2 K}{2}\mathbb{E}\left\|\frac{\sum_{i=1}^n \nabla f_i(\mathbf{x}_{i,t})}{n}\right\|^2
$$

$$
= \mathbb{E}f\left(\frac{X_t\mathbf{1}_n}{n}\right) - \frac{\alpha\varepsilon - \alpha^2\varepsilon^2 K}{2}\mathbb{E}\left\|\frac{\partial f(X_t)\mathbf{1}_n}{n}\right\|^2 - \frac{\alpha\varepsilon}{2}\mathbb{E}\left\|\nabla f\left(\frac{X_t\mathbf{1}_n}{n}\right)\right\|^2
$$

$$
+ \frac{\varepsilon^2 K}{2n}\sigma^2 + \frac{\alpha^2\varepsilon^2 K}{2n}\gamma_2^2
$$

$$
+ \frac{\alpha\varepsilon}{2}\underbrace{\mathbb{E}\left\|\nabla f\left(\frac{X_t\mathbf{1}_n}{n}\right) - \frac{\partial f(X_t)\mathbf{1}_n}{n}\right\|^2}_{T_1} \tag{39}
$$

where we used the identity $2\langle\mathbf{a}, \mathbf{b}\rangle = \|\mathbf{a}\|^2 + \|\mathbf{b}\|^2 - \|\mathbf{a} - \mathbf{b}\|^2$. The term $T_1$ defined in (39) can be bounded as follows:

$$
T_1 = \mathbb{E}\left\|\nabla f\left(\frac{X_t\mathbf{1}_n}{n}\right) - \frac{\partial f(X_t)\mathbf{1}_n}{n}\right\|^2
$$

$$
\leq \frac{1}{n}\sum_{i=1}^n \mathbb{E}\left\|\nabla f_i\left(\frac{X_t\mathbf{1}_n}{n}\right) - \nabla f_i(\mathbf{x}_{i,t})\right\|^2
$$

$$
\leq \frac{K^2}{n}\sum_{i=1}^n \mathbb{E}\underbrace{\left\|\frac{X_t\mathbf{1}_n}{n} - \mathbf{x}_{i,t}\right\|^2}_{Q_{i,t}}.
$$

Let us define

$$
Q_{i,t} := \mathbb{E}\left\|\frac{X_t\mathbf{1}_n}{n} - \mathbf{x}_{i,t}\right\|^2,
$$

and

$$
M_t := \frac{1}{n}\sum_{i=1}^n Q_{i,t} = \frac{1}{n}\sum_{i=1}^n \mathbb{E}\left\|\frac{X_t\mathbf{1}_n}{n} - \mathbf{x}_{i,t}\right\|^2.
$$

Here, $Q_{i,t}$ captures the deviation of the model at node $i$ from the average model at iteration $t$ and $M_t$ aggregates them to measure the average total consensus error. To bound $M_t$, we need to evaluate the following recursive expressions:

$$X_t = X_{t-1}W_\varepsilon + \varepsilon(Z_{t-1} - X_{t-1})(W - W_D) - \alpha\varepsilon\widetilde{\partial}f(X_{t-1})$$

$$= X_0 W_\varepsilon^t + \varepsilon\sum_{s=0}^{t-1}(Z_s - X_s)(W - W_D)W_\varepsilon^{t-s-1} - \alpha\varepsilon\sum_{s=0}^{t-1}\widetilde{\partial}f(X_s)W_\varepsilon^{t-s-1}. \qquad (40)$$

Now, using (40) we can write

$$M_t = \frac{1}{n}\sum_{i=1}^{n}\mathbb{E}\left\|\frac{X_t\mathbf{1}_n}{n} - \mathbf{x}_{i,t}\right\|^2$$

$$= \frac{1}{n}\mathbb{E}\|\bar{X}_t - X_t\|_F^2$$

$$= \frac{1}{n}\mathbb{E}\left\|X_t\frac{\mathbf{1}\mathbf{1}^\top}{n} - X_t\right\|_F^2$$

$$= \frac{1}{n}\mathbb{E}\left\|X_0\left(\frac{\mathbf{1}\mathbf{1}^\top}{n} - W_\varepsilon^t\right) + \varepsilon\sum_{s=0}^{t-1}(Z_s - X_s)(W - W_D)\left(\frac{\mathbf{1}\mathbf{1}^\top}{n} - W_\varepsilon^{t-s-1}\right)\right.$$

$$\left. - \alpha\varepsilon\sum_{s=0}^{t-1}\widetilde{\partial}f(X_s)\left(\frac{\mathbf{1}\mathbf{1}^\top}{n} - W_\varepsilon^{t-s-1}\right)\right\|_F^2$$

$$= \frac{(\alpha\varepsilon)^2}{n}\underbrace{\mathbb{E}\left\|\sum_{s=0}^{t-1}\widetilde{\partial}f(X_s)\left(\frac{\mathbf{1}\mathbf{1}^\top}{n} - W_\varepsilon^{t-s-1}\right)\right\|_F^2}_{T_2}$$

$$+ \frac{\varepsilon^2}{n}\underbrace{\mathbb{E}\left\|\sum_{s=0}^{t-1}(Z_s - X_s)(W - W_D)\left(\frac{\mathbf{1}\mathbf{1}^\top}{n} - W_\varepsilon^{t-s-1}\right)\right\|_F^2}_{T_3},$$

where we used the fact that quantiziations and stochastic gradients are statistically independent and $X_0 = 0$. We continue the analysis by bounding $T_2$ as follows:

$$T_2 = \mathbb{E}\left\|\sum_{s=0}^{t-1}\widetilde{\partial}f(X_s)\left(\frac{\mathbf{1}\mathbf{1}^\top}{n} - W_\varepsilon^{t-s-1}\right)\right\|_F^2$$

$$= \mathbb{E}\left\|\sum_{s=0}^{t-1}\left(\widetilde{\partial}f(X_s) - \partial f(X_s) + \partial f(X_s)\right)\left(\frac{\mathbf{1}\mathbf{1}^\top}{n} - W_\varepsilon^{t-s-1}\right)\right\|_F^2$$

$$\leq 2\underbrace{\mathbb{E}\left\|\sum_{s=0}^{t-1}\left(\widetilde{\partial}f(X_s) - \partial f(X_s)\right)\left(\frac{\mathbf{1}\mathbf{1}^\top}{n} - W_\varepsilon^{t-s-1}\right)\right\|_F^2}_{T_4}$$

$$+ 2\underbrace{\mathbb{E}\left\|\sum_{s=0}^{t-1}\partial f(X_s)\left(\frac{\mathbf{1}\mathbf{1}^\top}{n} - W_\varepsilon^{t-s-1}\right)\right\|_F^2}_{T_5}.$$

We can write

$$
\begin{aligned}
T_4 &= \mathbb{E}\left\|\sum_{s=0}^{t-1}\left(\widetilde{\partial}f(X_s) - \partial f(X_s)\right)\left(\frac{\mathbf{1}\mathbf{1}^\top}{n} - W_\varepsilon^{t-s-1}\right)\right\|_F^2 \\
&\leq \sum_{s=0}^{t-1}\mathbb{E}\left\|\widetilde{\partial}f(X_s) - \partial f(X_s)\right\|_F^2 \cdot \left\|\frac{\mathbf{1}\mathbf{1}^\top}{n} - W_\varepsilon^{t-s-1}\right\|^2 \\
&\leq n\gamma_2^2\sum_{s=0}^{t-1}\beta_\varepsilon^{2(t-s-1)} \\
&\leq \frac{n\gamma_2^2}{1 - \beta_\varepsilon^2},
\end{aligned}
\tag{41}
$$

where we used the facts that $\|AB\|_F \leq \|A\|_F\|B\|$ for matrices $A$, $B$ and also that $\left\|\frac{\mathbf{1}\mathbf{1}^\top}{n} - W_\varepsilon^t\right\| \leq \beta_\varepsilon^t$ for any $t = 0, 1, \cdots$. We continue by bounding $T_5$:

$$
\begin{aligned}
T_5 &= \mathbb{E}\left\|\sum_{s=0}^{t-1}\partial f(X_s)\left(\frac{\mathbf{1}\mathbf{1}^\top}{n} - W_\varepsilon^{t-s-1}\right)\right\|_F^2 \\
&= \underbrace{\sum_{s=0}^{t-1}\mathbb{E}\left\|\partial f(X_s)\left(\frac{\mathbf{1}\mathbf{1}^\top}{n} - W_\varepsilon^{t-s-1}\right)\right\|_F^2}_{T_6} \\
&\quad + \underbrace{\sum_{0\leq s\neq s'\leq t-1}\mathbb{E}\left\langle \partial f(X_s)\left(\frac{\mathbf{1}\mathbf{1}^\top}{n} - W_\varepsilon^{t-s-1}\right), \partial f(X_{s'})\left(\frac{\mathbf{1}\mathbf{1}^\top}{n} - W_\varepsilon^{t-s'-1}\right)\right\rangle_F}_{T_7}
\end{aligned}
\tag{42}
$$

Let us first bound the term $T_6$:

$$
\begin{aligned}
T_6 &= \sum_{s=0}^{t-1}\mathbb{E}\left\|\partial f(X_s)\left(\frac{\mathbf{1}\mathbf{1}^\top}{n} - W_\varepsilon^{t-s-1}\right)\right\|_F^2 \\
&\leq \sum_{s=0}^{t-1}\underbrace{\mathbb{E}\|\partial f(X_s)\|_F^2}_{T_8}\left\|\frac{\mathbf{1}\mathbf{1}^\top}{n} - W_\varepsilon^{t-s-1}\right\|^2,
\end{aligned}
\tag{43}
$$

where

$$
\begin{aligned}
T_8 &= \mathbb{E}\big\|\partial f(X_s)\big\|_F^2 \\
&\leq 3\mathbb{E}\left\|\partial f(X_s) - \partial f\left(\frac{X_s \mathbf{1}_n}{n}\mathbf{1}_n^\top\right)\right\|_F^2 \\
&\quad + 3\mathbb{E}\left\|\partial f\left(\frac{X_s \mathbf{1}_n}{n}\mathbf{1}_n^\top\right) - \nabla f\left(\frac{X_s \mathbf{1}_n}{n}\right)\mathbf{1}_n^\top\right\|_F^2 \\
&\quad + 3\mathbb{E}\left\|\nabla f\left(\frac{X_s \mathbf{1}_n}{n}\right)\mathbf{1}_n^\top\right\|_F^2 \\
&\leq 3\mathbb{E}\left\|\partial f(X_s) - \partial f\left(\frac{X_s \mathbf{1}_n}{n}\mathbf{1}_n^\top\right)\right\|_F^2 \\
&\quad + 3n\gamma_1^2 \\
&\quad + 3\mathbb{E}\left\|\nabla f\left(\frac{X_s \mathbf{1}_n}{n}\right)\mathbf{1}_n^\top\right\|_F^2 \\
&\leq 3K^2 \sum_{i=1}^n \mathbb{E}\left\|\frac{X_s \mathbf{1}_n}{n} - \mathbf{x}_{i,s}\right\|^2 + 3n\gamma_1^2 + 3\mathbb{E}\left\|\nabla f\left(\frac{X_s \mathbf{1}_n}{n}\right)\mathbf{1}_n^\top\right\|_F^2 \\
&= 3K^2 \sum_{i=1}^n Q_{i,s} + 3n\gamma_1^2 + 3\mathbb{E}\left\|\nabla f\left(\frac{X_s \mathbf{1}_n}{n}\right)\mathbf{1}_n^\top\right\|_F^2.
\end{aligned}
\tag{44}
$$

Plugging (44) in (43) yields

$$
\begin{aligned}
T_6 &\leq 3K^2 \sum_{s=0}^{t-1}\sum_{i=1}^n Q_{i,s}\left\|\frac{\mathbf{1}\mathbf{1}^\top}{n} - W_\varepsilon^{t-s-1}\right\|^2 \\
&\quad + 3n\gamma_1^2 \frac{1}{1-\beta_\varepsilon^2} \\
&\quad + 3\sum_{s=0}^{t-1}\mathbb{E}\left\|\nabla f\left(\frac{X_s \mathbf{1}_n}{n}\right)\mathbf{1}_n^\top\right\|_F^2\left\|\frac{\mathbf{1}\mathbf{1}^\top}{n} - W_\varepsilon^{t-s-1}\right\|^2
\end{aligned}
$$

Going back to terms $T_5$ and $T_7$, we can write

$$
\begin{aligned}
T_7 &= \sum_{s \neq s'}^{t-1} \mathbb{E} \left\langle \partial f(X_s) \left( \frac{\mathbf{1}\mathbf{1}^\top}{n} - W_\varepsilon^{t-s-1} \right), \partial f(X_{s'}) \left( \frac{\mathbf{1}\mathbf{1}^\top}{n} - W_\varepsilon^{t-s'-1} \right) \right\rangle_F \\
&\leq \sum_{s \neq s'}^{t-1} \mathbb{E} \left\| \partial f(X_s) \left( \frac{\mathbf{1}\mathbf{1}^\top}{n} - W_\varepsilon^{t-s-1} \right) \right\|_F \left\| \partial f(X_{s'}) \left( \frac{\mathbf{1}\mathbf{1}^\top}{n} - W_\varepsilon^{t-s'-1} \right) \right\|_F \\
&\leq \sum_{s \neq s'}^{t-1} \mathbb{E} \|\partial f(X_s)\|_F \left\| \frac{\mathbf{1}\mathbf{1}^\top}{n} - W_\varepsilon^{t-s-1} \right\| \|\partial f(X_{s'})\|_F \left\| \frac{\mathbf{1}\mathbf{1}^\top}{n} - W_\varepsilon^{t-s'-1} \right\| \\
&\leq \sum_{s \neq s'}^{t-1} \mathbb{E} \frac{\|\partial f(X_s)\|_F^2}{2} \left\| \frac{\mathbf{1}\mathbf{1}^\top}{n} - W_\varepsilon^{t-s-1} \right\| \left\| \frac{\mathbf{1}\mathbf{1}^\top}{n} - W_\varepsilon^{t-s'-1} \right\| \\
&\quad + \sum_{s \neq s'}^{t-1} \mathbb{E} \frac{\|\partial f(X_{s'})\|_F^2}{2} \left\| \frac{\mathbf{1}\mathbf{1}^\top}{n} - W_\varepsilon^{t-s-1} \right\| \left\| \frac{\mathbf{1}\mathbf{1}^\top}{n} - W_\varepsilon^{t-s'-1} \right\| \\
&\leq \sum_{s \neq s'}^{t-1} \mathbb{E} \left( \frac{\|\partial f(X_s)\|_F^2}{2} + \frac{\|\partial f(X_{s'})\|_F^2}{2} \right) \beta_\varepsilon^{2t-(s+s')-2} \\
&= \sum_{s \neq s'}^{t-1} \mathbb{E} \|\partial f(X_s)\|_F^2 \, \beta_\varepsilon^{2t-(s+s')-2} \\
&\leq 3 \underbrace{\sum_{s \neq s'}^{t-1} \left( 3K^2 \sum_{i=1}^n Q_{i,s} + 3\mathbb{E} \left\| \nabla f \left( \frac{X_s \mathbf{1}_n}{n} \right) \mathbf{1}_n^\top \right\|_F^2 \right) \beta_\varepsilon^{2t-(s+s')-2}}_{T_9} \\
&\quad + \underbrace{3n\gamma_1^2 \sum_{s \neq s'}^{t-1} \beta_\varepsilon^{2t-(s+s')-2}}_{T_{10}}.
\end{aligned}
$$

(45)

In above, the term $T_{10}$ can be simply bounded as:

$$
\begin{aligned}
T_{10} &= 3n\gamma_1^2 \sum_{s \neq s'}^{t-1} \beta_\varepsilon^{2t-(s+s')-2} \\
&= 6n\gamma_1^2 \sum_{s > s'}^{t-1} \beta_\varepsilon^{2t-(s+s')-2} \\
&= 6n\gamma_1^2 \frac{\left( \beta_\varepsilon^t - 1 \right) \left( \beta_\varepsilon^t - \beta_\varepsilon \right)}{\left( \beta_\varepsilon - 1 \right)^2 \left( \beta_\varepsilon + 1 \right)} \\
&\leq 6n\gamma_1^2 \frac{1}{\left( 1 - \beta_\varepsilon \right)^2}.
\end{aligned}
$$

The other term, i.e. $T_9$ can be bounded as follows:

$$
\begin{aligned}
T_9 &= 3 \sum_{s \neq s'}^{t-1} \left( 3K^2 \sum_{i=1}^{n} Q_{i,s} + 3\mathbb{E} \left\| \nabla f \left( \frac{X_s \mathbf{1}_n}{n} \right) \mathbf{1}_n^\top \right\|_F^2 \right) \beta_\varepsilon^{2t-(s+s')-2} \\
&= 6 \sum_{s=0}^{t-1} \left( 3K^2 \sum_{i=1}^{n} Q_{i,s} + 3\mathbb{E} \left\| \nabla f \left( \frac{X_s \mathbf{1}_n}{n} \right) \mathbf{1}_n^\top \right\|_F^2 \right) \sum_{s'=s+1}^{t-1} \beta_\varepsilon^{2t-(s+s')-2} \\
&\leq 6 \sum_{s=0}^{t-1} \left( 3K^2 \sum_{i=1}^{n} Q_{i,s} + 3\mathbb{E} \left\| \nabla f \left( \frac{X_s \mathbf{1}_n}{n} \right) \mathbf{1}_n^\top \right\|_F^2 \right) \frac{\beta_\varepsilon^{t-s-1}}{1 - \beta_\varepsilon}
\end{aligned}
$$

Now that we have bounded $T_6$ and $T_7$, we go back and plug in (42) to bound $T_5$:

$$
\begin{aligned}
T_5 &\leq 3K^2 \sum_{s=0}^{t-1} \sum_{i=1}^{n} Q_{i,s} \left\| \frac{\mathbf{11}^\top}{n} - W_\varepsilon^{t-s-1} \right\|^2 \\
&\quad + 3 \sum_{s=0}^{t-1} \mathbb{E} \left\| \nabla f \left( \frac{X_s \mathbf{1}_n}{n} \right) \mathbf{1}_n^\top \right\|_F^2 \left\| \frac{\mathbf{11}^\top}{n} - W_\varepsilon^{t-s-1} \right\|^2 \\
&\quad + 6 \sum_{s=0}^{t-1} \left( 3K^2 \sum_{i=1}^{n} Q_{i,s} + 3\mathbb{E} \left\| \nabla f \left( \frac{X_s \mathbf{1}_n}{n} \right) \mathbf{1}_n^\top \right\|_F^2 \right) \frac{\beta_\varepsilon^{t-s-1}}{1 - \beta_\varepsilon} \\
&\quad + 3n\gamma_1^2 \frac{1}{1 - \beta_\varepsilon^2} \\
&\quad + 6n\gamma_1^2 \frac{1}{(1 - \beta_\varepsilon)^2} \\
&\leq 3K^2 \sum_{s=0}^{t-1} \sum_{i=1}^{n} Q_{i,s} \left\| \frac{\mathbf{11}^\top}{n} - W_\varepsilon^{t-s-1} \right\|^2 \\
&\quad + 3 \sum_{s=0}^{t-1} \mathbb{E} \left\| \nabla f \left( \frac{X_s \mathbf{1}_n}{n} \right) \mathbf{1}_n^\top \right\|_F^2 \left\| \frac{\mathbf{11}^\top}{n} - W_\varepsilon^{t-s-1} \right\|^2 \\
&\quad + 6 \sum_{s=0}^{t-1} \left( 3K^2 \sum_{i=1}^{n} Q_{i,s} + 3\mathbb{E} \left\| \nabla f \left( \frac{X_s \mathbf{1}_n}{n} \right) \mathbf{1}_n^\top \right\|_F^2 \right) \frac{\beta_\varepsilon^{t-s-1}}{1 - \beta_\varepsilon} \\
&\quad + 9n\gamma_1^2 \frac{1}{(1 - \beta_\varepsilon)^2},
\end{aligned}
$$

where we used the fact that $\frac{1}{1-\beta_\varepsilon^2} \leq \frac{1}{(1-\beta_\varepsilon)^2}$. Now we bound the term $T_2$ having $T_4$ and $T_5$ bounded:

$$
\begin{aligned}
T_2 &= 2T_4 + 2T_5 \\
&\leq 2\frac{n\gamma_2^2}{1-\beta_\varepsilon^2} \\
&\quad + 6K^2 \sum_{s=0}^{t-1} \sum_{i=1}^{n} Q_{i,s} \left\| \frac{\mathbf{1}\mathbf{1}^\top}{n} - W_\varepsilon^{t-s-1} \right\|^2 \\
&\quad + 6 \sum_{s=0}^{t-1} \mathbb{E} \left\| \nabla f \left( \frac{X_s \mathbf{1}_n}{n} \right) \mathbf{1}_n^\top \right\|_F^2 \left\| \frac{\mathbf{1}\mathbf{1}^\top}{n} - W_\varepsilon^{t-s-1} \right\|^2 \\
&\quad + 12 \sum_{s=0}^{t-1} \left( 3K^2 \sum_{i=1}^{n} Q_{i,s} + 3\mathbb{E} \left\| \nabla f \left( \frac{X_s \mathbf{1}_n}{n} \right) \mathbf{1}_n^\top \right\|_F^2 \right) \frac{\beta_\varepsilon^{t-s-1}}{1-\beta_\varepsilon} \\
&\quad + 18n\gamma_1^2 \frac{1}{(1-\beta_\varepsilon)^2}.
\end{aligned}
$$

Moreover, the term $T_3$ can be bounded as follows:

$$
\begin{aligned}
T_3 &= \mathbb{E} \left\| \sum_{s=0}^{t-1} (Z_s - X_s)(W - W_D) \left( \frac{\mathbf{1}\mathbf{1}^\top}{n} - W_\varepsilon^{t-s-1} \right) \right\|_F^2 \\
&\leq \mathbb{E} \sum_{s=0}^{t-1} \|Z_s - X_s\|_F^2 \|W - W_D\|^2 \left\| \frac{\mathbf{1}\mathbf{1}^\top}{n} - W_\varepsilon^{t-s-1} \right\|^2 \\
&\leq \frac{4n\sigma^2}{1-\beta_\varepsilon^2},
\end{aligned}
$$

where we used the fact that $\|W - W_D\| \le 2$. Now we use the bounds derived for $T_2$ and $T_3$ to bound the consensus error $M_t$ as follows:

$$
\begin{aligned}
M_t &\le \frac{\alpha^2 \varepsilon^2}{n} T_2 + \frac{\varepsilon^2}{n} T_3 \\
&\le \frac{2\alpha^2 \varepsilon^2 \gamma_2^2}{1 - \beta_\varepsilon^2} \\
&\quad + \frac{6\alpha^2 \varepsilon^2 K^2}{n} \sum_{s=0}^{t-1} \sum_{i=1}^{n} Q_{i,s} \left\| \frac{\mathbf{1}\mathbf{1}^\top}{n} - W_\varepsilon^{t-s-1} \right\|^2 \\
&\quad + \frac{6\alpha^2 \varepsilon^2}{n} \sum_{s=0}^{t-1} \mathbb{E} \left\| \nabla f\left(\frac{X_s \mathbf{1}_n}{n}\right) \mathbf{1}_n^\top \right\|_F^2 \left\| \frac{\mathbf{1}\mathbf{1}^\top}{n} - W_\varepsilon^{t-s-1} \right\|^2 \\
&\quad + \frac{12\alpha^2 \varepsilon^2}{n} \sum_{s=0}^{t-1} \left( 3K^2 \sum_{i=1}^{n} Q_{i,s} + 3\mathbb{E} \left\| \nabla f\left(\frac{X_s \mathbf{1}_n}{n}\right) \mathbf{1}_n^\top \right\|_F^2 \right) \frac{\beta_\varepsilon^{t-s-1}}{1 - \beta_\varepsilon} \\
&\quad + \frac{18\alpha^2 \varepsilon^2 \gamma_1^2}{(1 - \beta_\varepsilon)^2} \\
&\quad + \frac{4\varepsilon^2 \sigma^2}{1 - \beta_\varepsilon^2} \\
&\le \frac{2\alpha^2 \varepsilon^2 \gamma_2^2}{1 - \beta_\varepsilon^2} + \frac{18\alpha^2 \varepsilon^2 \gamma_1^2}{(1 - \beta_\varepsilon)^2} + \frac{4\varepsilon^2 \sigma^2}{1 - \beta_\varepsilon^2} \\
&\quad + \frac{6\alpha^2 \varepsilon^2 K^2}{n} \sum_{s=0}^{t-1} \sum_{i=1}^{n} Q_{i,s} \beta_\varepsilon^{2(t-s-1)} \\
&\quad + \frac{6\alpha^2 \varepsilon^2}{n} \sum_{s=0}^{t-1} \mathbb{E} \left\| \nabla f\left(\frac{X_s \mathbf{1}_n}{n}\right) \mathbf{1}_n^\top \right\|_F^2 \beta_\varepsilon^{2(t-s-1)} \\
&\quad + \frac{12\alpha^2 \varepsilon^2}{n} \sum_{s=0}^{t-1} \left( 3K^2 \sum_{i=1}^{n} Q_{i,s} + 3\mathbb{E} \left\| \nabla f\left(\frac{X_s \mathbf{1}_n}{n}\right) \mathbf{1}_n^\top \right\|_F^2 \right) \frac{\beta_\varepsilon^{t-s-1}}{1 - \beta_\varepsilon} \\
&\le \frac{2\alpha^2 \varepsilon^2 \gamma_2^2}{1 - \beta_\varepsilon^2} + \frac{18\alpha^2 \varepsilon^2 \gamma_1^2}{(1 - \beta_\varepsilon)^2} + \frac{4\varepsilon^2 \sigma^2}{1 - \beta_\varepsilon^2} \\
&\quad + \frac{6\alpha^2 \varepsilon^2}{n} \sum_{s=0}^{t-1} \mathbb{E} \left\| \nabla f\left(\frac{X_s \mathbf{1}_n}{n}\right) \mathbf{1}_n^\top \right\|_F^2 \left( \beta_\varepsilon^{2(t-s-1)} + \frac{2\beta_\varepsilon^{t-s-1}}{1 - \beta_\varepsilon} \right) \\
&\quad + \frac{6\alpha^2 \varepsilon^2}{n} K^2 \sum_{s=0}^{t-1} \sum_{i=1}^{n} Q_{i,s} \left( \frac{2\beta_\varepsilon^{t-s-1}}{1 - \beta_\varepsilon} + \beta_\varepsilon^{2(t-s-1)} \right).
\end{aligned}
\tag{46}
$$

As we defined earlier, we have $M_s = \frac{1}{n} \sum_{i=1}^{n} Q_{i,s}$ which simplifies (46) to the following:

$$
\begin{aligned}
M_t &\le \frac{2\alpha^2 \varepsilon^2 \gamma_2^2}{1 - \beta_\varepsilon^2} + \frac{18\alpha^2 \varepsilon^2 \gamma_1^2}{(1 - \beta_\varepsilon)^2} + \frac{4\varepsilon^2 \sigma^2}{1 - \beta_\varepsilon^2} \\
&\quad + \frac{6\alpha^2 \varepsilon^2}{n} \sum_{s=0}^{t-1} \mathbb{E} \left\| \nabla f\left(\frac{X_s \mathbf{1}_n}{n}\right) \mathbf{1}_n^\top \right\|_F^2 \left( \beta_\varepsilon^{2(t-s-1)} + \frac{2\beta_\varepsilon^{t-s-1}}{1 - \beta_\varepsilon} \right) \\
&\quad + 6\alpha^2 \varepsilon^2 K^2 \sum_{s=0}^{t-1} M_s \left( \frac{2\beta_\varepsilon^{t-s-1}}{1 - \beta_\varepsilon} + \beta_\varepsilon^{2(t-s-1)} \right)
\end{aligned}
\tag{47}
$$

Now we can sum (47) over $t = 0, 1, \cdots, T-1$ which yields

$$
\begin{aligned}
\sum_{t=0}^{T-1} M_t &\leq \frac{2\alpha^2 \varepsilon^2 \gamma_2^2}{1 - \beta_\varepsilon^2} T + \frac{18\alpha^2 \varepsilon^2 \gamma_1^2}{(1 - \beta_\varepsilon)^2} T + \frac{4\varepsilon^2 \sigma^2}{1 - \beta_\varepsilon^2} T \\
&\quad + \frac{6\alpha^2 \varepsilon^2}{n} \sum_{t=0}^{T-1} \sum_{s=0}^{t-1} \mathbb{E} \left\| \nabla f \left( \frac{X_s \mathbf{1}_n}{n} \right) \mathbf{1}_n^\top \right\|_F^2 \left( \beta_\varepsilon^{2(t-s-1)} + \frac{2\beta_\varepsilon^{t-s-1}}{1 - \beta_\varepsilon} \right) \\
&\quad + 6\alpha^2 \varepsilon^2 K^2 \sum_{t=0}^{T-1} \sum_{s=0}^{t-1} M_s \left( \frac{2\beta_\varepsilon^{t-s-1}}{1 - \beta_\varepsilon} + \beta_\varepsilon^{2(t-s-1)} \right) \\
&\leq \frac{2\alpha^2 \varepsilon^2 \gamma_2^2}{1 - \beta_\varepsilon^2} T + \frac{18\alpha^2 \varepsilon^2 \gamma_1^2}{(1 - \beta_\varepsilon)^2} T + \frac{4\varepsilon^2 \sigma^2}{1 - \beta_\varepsilon^2} T \\
&\quad + \frac{6\alpha^2 \varepsilon^2}{n} \sum_{t=0}^{T-1} \mathbb{E} \left\| \nabla f \left( \frac{X_s \mathbf{1}_n}{n} \right) \mathbf{1}_n^\top \right\|_F^2 \left( \sum_{k=0}^{\infty} \beta_\varepsilon^{2k} + \frac{2\sum_{k=0}^{\infty} \beta_\varepsilon^k}{1 - \beta_\varepsilon} \right) \\
&\quad + 6\alpha^2 \varepsilon^2 K^2 \sum_{t=0}^{T-1} M_t \left( \frac{2\sum_{k=0}^{\infty} \beta_\varepsilon^k}{1 - \beta_\varepsilon} + \sum_{k=0}^{\infty} \beta_\varepsilon^{2k} \right) \\
&\leq \frac{2\alpha^2 \varepsilon^2 \gamma_2^2}{1 - \beta_\varepsilon^2} T + \frac{18\alpha^2 \varepsilon^2 \gamma_1^2}{(1 - \beta_\varepsilon)^2} T + \frac{4\varepsilon^2 \sigma^2}{1 - \beta_\varepsilon^2} T \\
&\quad + \frac{18\alpha^2 \varepsilon^2}{n(1 - \beta_\varepsilon)^2} \sum_{t=0}^{T-1} \mathbb{E} \left\| \nabla f \left( \frac{X_s \mathbf{1}_n}{n} \right) \mathbf{1}_n^\top \right\|_F^2 \\
&\quad + \frac{18\alpha^2 \varepsilon^2 K^2}{(1 - \beta_\varepsilon)^2} \sum_{t=0}^{T-1} M_t.
\end{aligned}
\tag{48}
$$

Note that $\left\| \nabla f \left( \frac{X_s \mathbf{1}_n}{n} \right) \mathbf{1}_n^\top \right\|_F^2 = n \left\| \nabla f \left( \frac{X_s \mathbf{1}_n}{n} \right) \right\|^2$, which simplifies (49) as follows:

$$
\begin{aligned}
\sum_{t=0}^{T-1} M_t &\leq \frac{2\alpha^2 \varepsilon^2 \gamma_2^2}{1 - \beta_\varepsilon^2} T + \frac{18\alpha^2 \varepsilon^2 \gamma_1^2}{(1 - \beta_\varepsilon)^2} T + \frac{4\varepsilon^2 \sigma^2}{1 - \beta_\varepsilon^2} T \\
&\quad + \frac{18\alpha^2 \varepsilon^2}{(1 - \beta_\varepsilon)^2} \sum_{t=0}^{T-1} \mathbb{E} \left\| \nabla f \left( \frac{X_s \mathbf{1}_n}{n} \right) \right\|^2 \\
&\quad + \frac{18\alpha^2 \varepsilon^2 K^2}{(1 - \beta_\varepsilon)^2} \sum_{t=0}^{T-1} M_t.
\end{aligned}
\tag{49}
$$

Rearranging the terms implies that

$$
\begin{aligned}
\left( 1 - \frac{18\alpha^2 \varepsilon^2 K^2}{(1 - \beta_\varepsilon)^2} \right) \sum_{t=0}^{T-1} M_t &\leq \frac{2\alpha^2 \varepsilon^2 \gamma_2^2}{1 - \beta_\varepsilon^2} T + \frac{18\alpha^2 \varepsilon^2 \gamma_1^2}{(1 - \beta_\varepsilon)^2} T + \frac{4\varepsilon^2 \sigma^2}{1 - \beta_\varepsilon^2} T \\
&\quad + \frac{18\alpha^2 \varepsilon^2}{(1 - \beta_\varepsilon)^2} \sum_{t=0}^{T-1} \mathbb{E} \left\| \nabla f \left( \frac{X_s \mathbf{1}_n}{n} \right) \right\|^2.
\end{aligned}
\tag{50}
$$

Now define

$$
D_2 := 1 - \frac{18\alpha^2 \varepsilon^2 K^2}{(1 - \beta_\varepsilon)^2},
$$

and rewrite (50) as

$$\sum_{t=0}^{T-1} M_t \leq \frac{2\alpha^2\varepsilon^2\gamma_2^2}{(1-\beta_\varepsilon^2)D_2}T + \frac{18\alpha^2\varepsilon^2\gamma_1^2}{(1-\beta_\varepsilon)^2D_2}T + \frac{4\varepsilon^2\sigma^2}{(1-\beta_\varepsilon^2)D_2}T$$
$$+ \frac{18\alpha^2\varepsilon^2}{(1-\beta_\varepsilon)^2D_2}\sum_{t=0}^{T-1}\mathbb{E}\left\|\nabla f\left(\frac{X_s\mathbf{1}_n}{n}\right)\right\|^2. \tag{51}$$

Note that from definition of $T_1$ we have

$$T_1 \leq \frac{K^2}{n}\sum_{i=1}^{n} Q_{i,t} = K^2 M_t.$$

Now use the above fact in the recursive equation (39) which we started with, that is

$$\mathbb{E}f\left(\frac{X_{t+1}\mathbf{1}_n}{n}\right) \leq \mathbb{E}f\left(\frac{X_t\mathbf{1}_n}{n}\right)$$
$$- \frac{\alpha\varepsilon - \alpha^2\varepsilon^2 K}{2}\mathbb{E}\left\|\frac{\partial f(X_t)\mathbf{1}_n}{n}\right\|^2 - \frac{\alpha\varepsilon}{2}\mathbb{E}\left\|\nabla f\left(\frac{X_t\mathbf{1}_n}{n}\right)\right\|^2$$
$$+ \frac{\varepsilon^2 K}{2n}\sigma^2 + \frac{\alpha^2\varepsilon^2 K}{2n}\gamma_2^2$$
$$+ \frac{\alpha\varepsilon K^2}{2}M_t. \tag{52}$$

If we sum (52) over $t = 0, 1, \cdots, T-1$, we get

$$\frac{\alpha\varepsilon - \alpha^2\varepsilon^2 K}{2}\sum_{t=0}^{T-1}\mathbb{E}\left\|\frac{\partial f(X_t)\mathbf{1}_n}{n}\right\|^2 + \frac{\alpha\varepsilon}{2}\sum_{t=0}^{T-1}\mathbb{E}\left\|\nabla f\left(\frac{X_t\mathbf{1}_n}{n}\right)\right\|^2$$
$$\leq f(0) - f^* + \frac{\varepsilon^2 K}{2n}\sigma^2 T + \frac{\alpha^2\varepsilon^2 K}{2n}\gamma_2^2 T$$
$$+ \frac{\alpha\varepsilon K^2}{2}\sum_{t=0}^{T-1}M_t$$
$$\overset{\text{from (51)}}{\leq} f(0) - f^* + \frac{\varepsilon^2 K}{2n}\sigma^2 T + \frac{\alpha^2\varepsilon^2 K}{2n}\gamma_2^2 T$$
$$+ \frac{\alpha\varepsilon K^2}{2}\left\{\frac{2\alpha^2\varepsilon^2\gamma_2^2}{(1-\beta_\varepsilon^2)D_2}T + \frac{18\alpha^2\varepsilon^2\gamma_1^2}{(1-\beta_\varepsilon)^2D_2}T + \frac{4\varepsilon^2\sigma^2}{(1-\beta_\varepsilon^2)D_2}T\right\}$$
$$+ \frac{9\alpha^3\varepsilon^3 K^2}{(1-\beta_\varepsilon)^2D_2}\sum_{t=0}^{T-1}\mathbb{E}\left\|\nabla f\left(\frac{X_s\mathbf{1}_n}{n}\right)\right\|^2. \tag{53}$$

We ca rearrange the terms in (53) and rewrite it as

$$\frac{\alpha\varepsilon - \alpha^2\varepsilon^2 K}{2}\sum_{t=0}^{T-1}\mathbb{E}\left\|\frac{\partial f(X_t)\mathbf{1}_n}{n}\right\|^2 + \alpha\varepsilon\left(\frac{1}{2} - \frac{9\alpha^2\varepsilon^2 K^2}{(1-\beta_\varepsilon)^2D_2}\right)\sum_{t=0}^{T-1}\mathbb{E}\left\|\nabla f\left(\frac{X_t\mathbf{1}_n}{n}\right)\right\|^2$$
$$\leq f(0) - f^* + \frac{\varepsilon^2 K}{2n}\sigma^2 T + \frac{\alpha^2\varepsilon^2 K}{2n}\gamma_2^2 T$$
$$+ \frac{\alpha\varepsilon K^2}{2}\left\{\frac{2\alpha^2\varepsilon^2\gamma_2^2}{(1-\beta_\varepsilon^2)D_2}T + \frac{18\alpha^2\varepsilon^2\gamma_1^2}{(1-\beta_\varepsilon)^2D_2}T + \frac{4\varepsilon^2\sigma^2}{(1-\beta_\varepsilon^2)D_2}T\right\} \tag{54}$$

Now, we define $D_1$ as follows

$$D_1 := \frac{1}{2} - \frac{9\alpha^2\varepsilon^2 K^2}{(1-\beta_\varepsilon)^2D_2},$$

and replace in (54) which yields

$$\frac{1}{\alpha\varepsilon T}\left\{\frac{\alpha\varepsilon - \alpha^2\varepsilon^2 K}{2}\sum_{t=0}^{T-1}\mathbb{E}\left\|\frac{\partial f(X_t)\mathbf{1}_n}{n}\right\|^2 + \alpha\varepsilon D_1 \sum_{t=0}^{T-1}\mathbb{E}\left\|\nabla f\left(\frac{X_t\mathbf{1}_n}{n}\right)\right\|^2\right\}$$

$$\leq \frac{1}{\alpha\varepsilon T}(f(0) - f^*) + \frac{\varepsilon}{\alpha}\frac{K\sigma^2}{2n} + \alpha\varepsilon\frac{K\gamma_2^2}{2n}$$

$$+ \frac{\alpha^2\varepsilon^2}{1-\beta_\varepsilon^2}\frac{K^2\gamma_2^2}{D_2} + \frac{\alpha^2\varepsilon^2}{(1-\beta_\varepsilon)^2}\frac{9K^2\gamma_1^2}{D_2} + \frac{\varepsilon^2}{1-\beta_\varepsilon^2}\frac{2K^2\sigma^2}{D_2}. \tag{55}$$

To balance the terms in RHS of (55), we need to know how $\beta_\varepsilon$ behaves with $\varepsilon$. As we defined before, $W_\varepsilon = (1-\varepsilon)I + \varepsilon W$. Hence, $\lambda_i(W_\varepsilon) = 1 - \varepsilon + \varepsilon\lambda_i(W)$. Therefore, for $\varepsilon \leq \frac{1}{1-\lambda_n(W)}$, we have

$$\beta_\varepsilon = \max\left\{|\lambda_2(W_\varepsilon)|, |\lambda_n(W_\varepsilon)|\right\}$$
$$= \max\left\{|1 - \varepsilon + \varepsilon\lambda_2(W)|, |1 - \varepsilon + \varepsilon\lambda_n(W)|\right\}$$
$$= \max\left\{1 - \varepsilon + \varepsilon\lambda_2(W), 1 - \varepsilon + \varepsilon\lambda_n(W)\right\}$$
$$= 1 - \varepsilon\left(1 - \lambda_2(W)\right).$$

Therefore,

$$1 - \beta_\varepsilon = \varepsilon\left(1 - \lambda_2(W)\right) \geq \varepsilon(1-\beta)$$
$$1 - \beta_\varepsilon^2 = 2\varepsilon\left(1 - \lambda_2(W)\right) - \varepsilon^2\left(1 - \lambda_2(W)\right)^2 \geq \varepsilon(1-\beta^2).$$

Moreover, if $\alpha\varepsilon \leq \frac{1}{K}$ we have from (55) that

$$\frac{D_1}{T}\sum_{t=0}^{T-1}\mathbb{E}\left\|\nabla f\left(\frac{X_t\mathbf{1}_n}{n}\right)\right\|^2 \leq \frac{1}{\alpha\varepsilon T}(f(0) - f^*) + \frac{\varepsilon}{\alpha}\frac{K\sigma^2}{2n} + \alpha\varepsilon\frac{K\gamma_2^2}{2n}$$

$$+ \frac{\alpha^2\varepsilon}{1-\beta^2}\frac{K^2\gamma_2^2}{D_2} + \frac{\alpha^2}{(1-\beta)^2}\frac{9K^2\gamma_1^2}{D_2} + \frac{\varepsilon}{1-\beta^2}\frac{2K^2\sigma^2}{D_2}. \tag{56}$$

For $\alpha \leq \frac{1-\beta}{6K}$ we have

$$D_2 = 1 - \frac{18\alpha^2\varepsilon^2 K^2}{(1-\beta_\varepsilon)^2}$$
$$= 1 - \frac{18\alpha^2\varepsilon^2 K^2}{\varepsilon^2(1-\beta)^2}$$
$$= 1 - \frac{18\alpha^2 K^2}{(1-\beta)^2}$$
$$\geq \frac{1}{2}, \tag{57}$$

and for $\alpha \leq \frac{1-\beta}{6\sqrt{2}K}$ we have

$$D_1 = \frac{1}{2} - \frac{9\alpha^2\varepsilon^2 K^2}{(1-\beta_\varepsilon)^2 D_2}$$
$$\geq \frac{1}{2} - \frac{18\alpha^2\varepsilon^2 K^2}{\varepsilon^2(1-\beta)^2}$$
$$= \frac{1}{2} - \frac{18\alpha^2 K^2}{(1-\beta)^2}$$
$$\geq \frac{1}{4}.$$

Now, we pick the step-sizes as follows:

$$\alpha = \frac{1}{T^{1/6}}, \tag{58}$$

$$\varepsilon = \frac{1}{T^{1/2}}. \tag{59}$$

It is clear that in order to satisfy the conditions mentioned before, that are $\varepsilon \leq \frac{1}{1-\lambda_n(W)}$, $\alpha\varepsilon \leq \frac{1}{K}$ and $\alpha \leq \frac{1-\beta}{6\sqrt{2}K}$, it suffices to pick $T$ as large as the following:

$$T \geq T_{\min}^{\text{nc}} := \max\left\{(1-\lambda_n(W))^2, K^{3/2}, \left(\frac{6\sqrt{2}K}{1-\beta}\right)^6\right\}. \tag{60}$$

For such $T$ we have

$$
\begin{aligned}
\frac{1}{T}\sum_{t=0}^{T-1}\mathbb{E}\left\|\nabla f\left(\frac{X_t\mathbf{1}_n}{n}\right)\right\|^2 &\leq \frac{1}{T^{1/3}}4(f(0)-f^*) + \frac{1}{T^{1/3}}\frac{2K\sigma^2}{n} + \frac{1}{T^{2/3}}\frac{2K\gamma_2^2}{n}\\
&\quad + \frac{1}{T^{5/6}}\frac{8K^2\gamma_2^2}{1-\beta^2} + \frac{1}{T^{1/3}}\frac{72K^2\gamma_1^2}{(1-\beta)^2} + \frac{1}{T^{1/2}}\frac{16K^2\sigma^2}{1-\beta^2}\\
&= \frac{B_1}{T^{1/3}} + \frac{B_2}{T^{1/2}} + \frac{B_3}{T^{2/3}} + \frac{B_4}{T^{5/6}} \tag{61}\\
&= \mathcal{O}\left(\frac{K^2}{(1-\beta)^2}\frac{\gamma^2}{m} + K\frac{\sigma^2}{n}\right)\frac{1}{T^{1/3}}\\
&\quad + \mathcal{O}\left(\frac{K^2}{1-\beta^2}\sigma^2\right)\frac{1}{T^{1/2}}\\
&\quad + \mathcal{O}\left(K\frac{\gamma^2}{n}\max\left\{\frac{\mathbb{E}[1/V]}{T_d},\frac{1}{m}\right\}\right)\frac{1}{T^{2/3}}\\
&\quad + \mathcal{O}\left(\frac{K^2}{1-\beta^2}\gamma^2\max\left\{\frac{\mathbb{E}[1/V]}{T_d},\frac{1}{m}\right\}\right)\frac{1}{T^{5/6}},
\end{aligned}
$$

where

$$B_1 := 4(f(0)-f^*) + \frac{72K^2\gamma_1^2}{(1-\beta)^2} + \frac{2K\sigma^2}{n}$$

$$B_2 := \frac{16K^2\sigma^2}{1-\beta^2}$$

$$B_3 := \frac{2K\gamma_2^2}{n}$$

$$B_4 := \frac{8K^2\gamma_2^2}{1-\beta^2}.$$

Now we bound the consensus error. From (51) we have

$$
\begin{aligned}
\frac{1}{T}\sum_{t=0}^{T-1}\frac{1}{n}\sum_{i=1}^{n}\mathbb{E}\left\|\frac{X_t\mathbf{1}_n}{n}-\mathbf{x}_{i,t}\right\|^2 &= \frac{1}{T}\sum_{t=0}^{T-1}M_t\\
&\leq \frac{2\alpha^2\varepsilon^2\gamma_2^2}{(1-\beta_\varepsilon^2)D_2} + \frac{18\alpha^2\varepsilon^2\gamma_1^2}{(1-\beta_\varepsilon)^2 D_2} + \frac{4\varepsilon^2\sigma^2}{(1-\beta_\varepsilon^2)D_2}\\
&\quad + \frac{18\alpha^2\varepsilon^2}{(1-\beta_\varepsilon)^2 D_2}\frac{1}{T}\sum_{t=0}^{T-1}\mathbb{E}\left\|\nabla f\left(\frac{X_s\mathbf{1}_n}{n}\right)\right\|^2\\
&\leq \alpha^2\varepsilon\frac{2\gamma_2^2}{(1-\beta^2)D_2} + \alpha^2\frac{18\gamma_1^2}{(1-\beta)^2 D_2}\\
&\quad + \varepsilon\frac{4\sigma^2}{(1-\beta^2)D_2}\\
&\quad + \alpha^2\frac{18}{(1-\beta)^2 D_2}\frac{1}{T}\sum_{t=0}^{T-1}\mathbb{E}\left\|\nabla f\left(\frac{X_t\mathbf{1}_n}{n}\right)\right\|^2
\end{aligned}
$$

For the same step-sizes $\alpha$ and $\varepsilon$ defined in (58) and large enough $T$ as in (60), we can use the convergence result in (61) which yields

$$\frac{1}{T}\sum_{t=0}^{T-1}\frac{1}{n}\sum_{i=1}^{n}\mathbb{E}\left\|\frac{X_t\mathbf{1}_n}{n}-\mathbf{x}_{i,t}\right\|^2 \leq \frac{1}{T^{5/6}}\frac{4\gamma_2^2}{1-\beta^2}+\frac{1}{T^{1/3}}\frac{36\gamma_1^2}{(1-\beta)^2}+\frac{1}{T^{1/2}}\frac{8\sigma^2}{1-\beta^2}$$

$$+\frac{1}{T^{1/3}}\frac{36}{(1-\beta)^2}\left(\frac{B_1}{T^{1/3}}+\frac{B_2}{T^{1/2}}+\frac{B_3}{T^{2/3}}+\frac{B_4}{T^{5/6}}\right)$$

$$=\frac{C_1}{T^{1/3}}+\frac{C_2}{T^{1/2}}+\frac{C_3}{T^{2/3}}+\frac{C_4}{T^{5/6}}+\frac{C_5}{T}+\frac{C_6}{T^{7/6}}$$

$$=\mathcal{O}\left(\frac{\gamma^2}{m(1-\beta)^2}\right)\frac{1}{T^{1/3}}$$

$$+\mathcal{O}\left(\frac{\sigma^2}{1-\beta^2}\right)\frac{1}{T^{1/2}}$$

$$+\mathcal{O}\left(\frac{K^2}{(1-\beta)^4}\frac{\gamma^2}{m}+\frac{K}{(1-\beta)^2}\frac{\sigma^2}{n}\right)\frac{1}{T^{2/3}}$$

$$+\mathcal{O}\left(\frac{\gamma^2}{1-\beta^2}\max\left\{\frac{\mathbb{E}[1/V]}{T_d},\frac{1}{m}\right\}+\frac{K^2\sigma^2}{(1-\beta)^4}\right)\frac{1}{T^{5/6}}$$

$$+\mathcal{O}\left(\frac{K}{(1-\beta)^2}\frac{\gamma^2}{n}\max\left\{\frac{\mathbb{E}[1/V]}{T_d},\frac{1}{m}\right\}\right)\frac{1}{T}$$

$$+\mathcal{O}\left(\frac{K^2}{(1-\beta)^4}\gamma^2\max\left\{\frac{\mathbb{E}[1/V]}{T_d},\frac{1}{m}\right\}\right)\frac{1}{T^{7/6}},$$

where

$$C_1 := \frac{36\gamma_1^2}{(1-\beta)^2}$$

$$C_2 := \frac{8\sigma^2}{1-\beta^2}$$

$$C_3 := \frac{36}{(1-\beta)^2}B_1$$

$$C_4 := \frac{4\gamma_2^2}{1-\beta^2}+\frac{36}{(1-\beta)^2}B_2$$

$$C_5 := \frac{36}{(1-\beta)^2}B_3$$

$$C_6 := \frac{36}{(1-\beta)^2}B_4.$$