[Reviews · NeurIPS 2019]

Reviewer 1



This paper propose a new algorithm to solve a stochastic optimization problem distributively over the nodes of a graph of computing units. As usual, the algorithm involves a communication step and a computing step at each iteration. The algorithms only involves local computations. To be robust to the effect of stragglers among the computing units, the proposed algorithm imposes a deadline in computation time at each iteration. To avoid the communication cost, only quantized version of the local iterates are exchanged during communication steps. Two theorems are given : Th 1. The loss is strongly convex and a bound is provided for the expected square distance to the solution. Th 2. The loss is non convex and a bound is provided for the expected square norm of the gradient **at the average iterate**. A bound is also provided for the distance to the consensus. The problem is relevant. Deadlines, local computations and quantizations has been considered in the literature of collaborative learning but separately. The explanations are a bit technical but clear and the paper is well written. Numerical simulations are provided. I have two comments for the authors. 1. Assumption 2. It is assumed that the variance of the quantization is bounded whereas some authors allow it to grow with \|x\|. Could you remove this assumption? 2. Eq 10 should include the cost of computing the average iterate since in this paper, communication time is a bottleneck. *********************************************************************************** I appreciate the authors's response, especially the considerations for the convex setting and for the variance of the quantization (not for the convex setting, of course)

Reviewer 2



I have read the authors’ response. Overall, I think that this is a solid paper with significant theoretical and algorithmic contribution, and it is also well-written. A minor concern is that this paper established the convergence results under strongly convex and non-convex settings, whether the result can be directly derived for convex setting, i.e., the case of \mu=0 in Section 4.1. I hope that the results for convex setting can be added in the final version.

Reviewer 3



This paper combines gradient with deadline and quantization techniques into the decentralized optimization algorithm. Although the convergence in the nonconvex setting looks interesting and solid, the combination limits the novelty of this paper, 1. The experiments are too simple to show the efficacy of the proposed algorithms, merely MNIST and CIFAR10 datasets. 2. The size of the tested neural network is too small. In addition, whats the activation function for the fully connected neural network? If the ReLU activation was used, the tested loss function is not smooth. 3. For first order stochastic methods, their performances are sensitive to the learning rate. The authors should report the best results for each solver with well-tuned learning rates

[Author Response · NeurIPS 2019]

We thank the reviewers for their careful consideration and their feedback, our short replies are provided below.

**General response for Reviewers 1 & 2.** -*"Can the results be extended to convex loss functions?"*.

**Response:** We studied the convergence analysis of our proposed method for strongly-convex and non-convex settings
in the paper. Indeed, we can extend our results to the convex case by choosing the stepsizes $(\alpha, \varepsilon) = (T^{-\delta/4}, T^{-3\delta/4})$
which will lead to a sublinear rate of $f(\frac{1}{T}\sum_{t=1}^T \bar{\mathbf{x}}_t) - f^* \leq \mathcal{O}(T^{-\delta/2})$ for any $\delta \in (0, 1/2)$. Here is a sketch of the
proof. Note that our approach to prove the convergence in the strongly-convex case was two-folded: to show that (1) the
sequence of our method $\mathbf{x}_t$ converges to the optimizer of the penalty function $\mathbf{x}_\alpha^*$; and (2) $\mathbf{x}_\alpha^*$ converges to the global
optimizer $\mathbf{x}^*$. Similarly in eq. (20) but for convex loss, we have $2(h_\alpha(\mathbf{x}_t) - h_\alpha^*) \leq \varepsilon^{-1}\mathbb{E}\|\mathbf{x}_t - \mathbf{x}_\alpha^*\|^2 - \varepsilon^{-1}\mathbb{E}\|\mathbf{x}_{t+1} -$
$\mathbf{x}_\alpha^*\|^2 + \varepsilon\mathbb{E}\|\widetilde{\nabla} h_\alpha(\mathbf{x}_t)\|^2$. Together with eq. (21), we can simplify the previous telescopic sum and conclude the
convergence of $h_\alpha(\frac{1}{T}\sum_{t=1}^T \mathbf{x}_t) - h_\alpha^*$. Moreover, for the picked stepsizes, we can use the proof of standard SGD for
convex losses and show that $\alpha^{-1}h_\alpha^* \to f^*$, as well $\alpha^{-1}h_\alpha(\frac{1}{T}\sum_{t=1}^T \mathbf{x}_t) \to f(\frac{1}{T}\sum_{t=1}^T \bar{\mathbf{x}}_t)$, all at rate $\mathcal{O}(T^{-\delta/2})$.

**Reviewer 1.** -*"Could you remove the assumption that variance of quantization $\leq \sigma^2$ and allow it grow with $\|\mathbf{x}\|$?"*.

**Response:** Yes! For both strongly-convex and non-convex losses, we can assume that $\mathbb{E}\|Q(\mathbf{x}) - \mathbf{x}\|^2 \leq \sigma^2\|\mathbf{x}\|^2$ and
modify the proof as follows. For strongly-convex, in eq. (21) we'll have $\mathbb{E}\|\mathbf{z}_t - \mathbf{x}_t\|^2 \leq \sigma^2\|\mathbf{x}_t\|^2 \leq 2\sigma^2\|\mathbf{x}_t - \mathbf{x}_\alpha^*\|^2 +$
$2\sigma^2\|\mathbf{x}_\alpha^*\|^2$. The first term $\|\mathbf{x}_t - \mathbf{x}_\alpha^*\|^2$ can be simply merged in eq. (22); and the second term can be bounded as
$\|\mathbf{x}_\alpha^*\|^2 \leq 2\|\mathbf{x}_\alpha^* - \mathbf{x}^*\|^2 + 2\|\mathbf{x}^*\|^2$ where $\|\mathbf{x}_\alpha^* - \mathbf{x}^*\|^2$ decays by $\mathcal{O}(T^{-\delta})$ and $\|\mathbf{x}^*\|^2 \leq 2(f(0) - f^*)/\mu$. For non-convex
settings, we can follow the proof in the paper and replace $\mathbb{E}\|\mathbf{z}_t - \mathbf{x}_t\|^2 \leq \sigma^2\|\mathbf{x}_t\|^2 \leq 2\sigma^2\|\mathbf{x}_t - \bar{\mathbf{x}}_t\|^2 + 2\sigma^2\|\bar{\mathbf{x}}_t\|^2$ in
eq. (37). The first term is consensus error and will be merged in $T_1$ in eq. (39). The second term can be bounded
by noting that the function value decreases at each iteration and considering the typical assumption that $f(\mathbf{x}) \to \infty$
when $\|\mathbf{x}\| \to \infty$. On the other hand, quantizers satisfying $\mathbb{E}\|Q(\mathbf{x}) - \mathbf{x}\|^2 \leq \sigma^2$ are indeed common in both theory and
practice; e.g., the low-precision quantizer, randomly rounding operator, quantization sparsifier studied in ref. [61].

-*"Eq 10 should include the cost of computing the average iterate since in this paper, communication time is a bottleneck."*.
**Response:** Computing the average iterate contains vector-scalar multiplication and vector-vector addition; however, it
is known in systems literature that these are negligible and the dominant computation cost/time is induced by matrix
multiplication (e.g. gradient computation for least-squares) which we have modeled in the paper. Moreover, eq. (10)
characterizes the convergence rate vs. iterations (and not wall-clock time).

-*"Explain the difference between deadline-based & asynchronous"*.
**Response:** The top figure schematically shows the differences between
deadline-based and asynchronous methods. In particular, in Asynchronous
DSGD, each worker continuously updates its local model according to the
most recent models of its neighbours, while in our deadline-based method,
each worker computes a batched gradient by the deadline $T_d$ (with a
random size depending on the speed) and then updates synchronously with
other workers. We will add this discussion in the revised paper.

**Reviewer 3.** -*"Experiments are too simple to show the efficacy of the*
*method, merely MNIST/CIFAR10. Size of the neural network is too small."*.
**Response:** We have conducted more experiments over the ImageNet
dataset which is known as a complicated dataset. As the middle figure
demonstrates, for a binary classification, our method significantly im-
proves upon the benchmarks over this dataset as well. We also carried
out experiments on a deeper neural network with 4 hidden layers and
our method provides significant speedups over the benchmarks (bottom
figure). This further demonstrates that our method is less sensitive to the
dataset or the neural network. Nevertheless, we would like to highlight that
our focus in this paper is to develop the *theory* of a provably converging,
straggler-resilient and communication-efficient framework.

-*"Whats the activation function for the fully connected neural network?*
*If the ReLU activation was used, the tested loss function is not smooth."*
**Response:** In all experiments, a sigmoid function is used as the activation
function for the neural network which makes the loss function smooth and
hence compatible with the theory. Extending our results to the nonsmooth
losses (e.g. ReLU) is our future direction.

-*"First order stochastic methods are sensitive to learning rate. The authors should report results with well-tuned rates."*.
**Response:** All the numerical results in our original submission indeed correspond to well-tuned learning rates; we will
highlight this point in the revised paper.

[Meta-Review · NeurIPS 2019]

This paper considers distributed solving of collaborative learning problem in the presence of stragglers and communication cost. The two main ideas are quantization of gradients and time limit on local gradient computation. Overall, this paper considers an important problem and solves two main issues in its implementation.